# PROOF-RM: A SCALABLE AND GENERALIZABLE REWARD MODEL FOR MATH PROOF

## ABSTRACT

Large Language Models (LLMs) have demonstrated strong math reasoning abilities through Reinforcement Learning with *Verifiable Rewards* (RLVR), where correctness can be assessed and rewarded by directly comparing answers. However, many challenging problems like Olympiad tasks or genuine mathematical research are proof-based, with no guaranteed way to determine the authenticity of a proof by simple answer matching. To enable scalable and efficient reinforcement learning in this setting, a Reward Model (RM) capable of reliably evaluating full proof processes is required. In this work, we design a *scalable* data-construction pipeline that, with minimal human effort, leverages LLMs to generate a large quantity of high-quality and diverse "**question-proof-check**" triplet data. These triplets train a proof-checking reward model under RLVR by rewarding correct verification outcomes. By varying problem sources, proof-generation methods, prompts, and generating LLMs, we obtain problem–proof pairs—both correct and non-trivially incorrect—that span difficulty, length, linguistic style, and error type. A hierarchical human review then filters for LLM–human aligned check labels used in training. With this data generation process, we train a proof RM that can accurately judge across diverse datasets, where additional process reward and token weight balance are used to stable long-term reward RL. Our experiments validate the model's scalability and effectiveness from multiple perspectives, including reward accuracy and test-time guidance, providing important practical recipes and tools for strengthening LLM mathematical capabilities.

## 1 INTRODUCTION

Pretrained large language models (LLMs) have acquired the ability to produce complex Chain-of-Thought (CoT) through reinforcement learning-based post-training, enabling success on increasing difficult tasks, such as mathematical problems solving (DeepSeek, 2025; Huang & Yang, 2025). Reinforcement Learning relies on *accurate* and *comprehensive* reward signals to ensure stable and efficient training and to avoid reward hacking (Amodei et al., 2016). Even a small amount of imprecise rewards can impede learning (Gao et al., 2024; Xu et al., 2025) or steer model towards shortcut reasoning (Gao et al., 2022; Pang et al., 2023), substantially degrading reasoning performance. Meanwhile, RL involves evaluating highly diverse, on-policy data generated online by the model. A reward system must possess both *generalizability* and *accuracy*.

The classic Reinforcement Learning from Human Feedback (RLHF) pipeline collects human preference data and train a neural Reward Model (RM) to approximate human favor. While RLHF has been successful in scenarios such as chatbot and creative writing, it is constrained by the scope and quantity of human-annotations for complex reasoning, where expert labeling is extremely costly. According to Dekoninck et al. (2025), in Olympic-level math, annotating a single question requires domain experts (like IMO participants) to spend hours and costs hundreds of dollars. In addition, reasoning tasks often involve rich solution paths and vast search spaces, demanding more diverse supervision to achieve reward generalization.

Reinforcement Learning with Verifiable Rewards (RLVR), proposed by Gao et al. (2024) and further exemplified by Tulu-3 (Lambert et al., 2025) and DeepSeek-R1 (DeepSeek-AI, 2025), addresses this challenge by providing a verifiable reward signal. RLVR leverages the "**asymmetry of verification**" (Wei, 2025), that is, in domains such as math, coding, web operations and so on,*solving

*a problem may be complex, but verifying an answer can be very simple*. In many math datasets and benchmarks, final answers are often in the forms of options, numerical values, or short expressions (see Table 1). Such instances are *"verifiable problems"*: one can directly compare a model-generated answer with the groundtruth to determine correctness. This enables accurate, scalable reward across a wide range of problems and thereby facilitating efficient large-scale RL. Consequently, mainstream post-training practices continue to adopt this approach (Xu et al., 2025; Chen et al., 2025a; Jimenez et al., 2024).

However, easy verifiability does not always hold. Verification asymmetry depends on the task's *result space*, *expression format*, and *evaluation criteria*. The current successes of RLVR heavily rely on the settings where the asymmetry is especially strong (Su et al., 2025; Huang et al., 2025). Multiple-choice questions exhibit the strongest asymmetry. Fill-in-the-blank questions already require additional judgment about the equivalence of mathematical expressions. For full proofs, verification asymmetry largely diminishes. To determine a proof is correct, at least the following requirements should be checked: adopting the 1) correct and 2) useful theorems, 3) making sure the conditions of theorems are satisfied, 4) arranging them correctly to get the target conclusion. It requires step-by-step careful checking and an overall understanding of the logic flow.

Unfortunately, in top-competition-level and research-level math, problems are predominantly proof-based. For example, in USAMO 2025 [1], 4 out of 6 problems are proof problem and the other 2 require both an answer and the proof. Considering that costly expert checking is now the only way to accurately evaluate these proof (Petrov et al., 2025), there is an increasingly urgent need to have a model that can verify the correctness of proofs. Because it determine the correctness of the proof and work as an RM, in the following part of the paper, we call the model *to read the question (Q), check the proof (P) and return the check (C) results as "ture/false" (T/F)* as **Proof RM**.

Training a proof RM requires QPC triplets with both correct and incorrect proofs. The proofs in real world are *flexible* and *heterogeneous*: for the same problem, valid proofs can differ substantially in logical flow, pivotal steps, linguistic style, and level of detail. Incorrect proofs further exhibit a wide spectrum of *failure modes* from incomplete proofs, logical gaps to unaddressed corner cases and even deliberate *"hand-wavy* bluff proofs". Consequently, the training corpus must be sufficiently diverse to cover as much of the proof space as possible in order to learn a generalizable proof RM. In contrast, existing QP corpora are often insufficiently diverse and imbalanced, which are often compiled from official solutions and are therefore concise. More critically, corpora of incorrect proofs, particularly non-trivial ones, are scarcely available or annotated.

In this paper, we introduce a data-collection pipeline that combines real-world question-proof pairs with "*LLM-aided multi-dimensional diversity expansion*" (Step 1 in Figure 1). Starting from some seed questions and human-written proof collected from the official solutions or teacher-verified student scripts across diverse sources and difficulty level, we further extend diversity by prompting LLMs to modify existing proofs and generated new proofs for these questions. We find that varying input format and prompt methods yields proofs with increased diversity on language style, proof verbosity and non-trivial **error types of incorrect proof**. Details are described in Section 3.1.

To scale the T/F annotations (Step 2 in Figure 1), we employ three LLM that judge each proof with a total of five times. The label will be accepted only under **unanimous** agreement. Because even all unanimous LLMs can still be wrong on hard problems but human checking is expensive, we operate a *combination*-level sampling check to minimal human efforts. We partition data into different combinations by its data source, prompt method, and generating LLM, like "C1", "C2" in Figure 1. For each combination , if sampled human judgments align strongly with LLMs, we treat that slice as silver-standard and keep them; otherwise we drop the entire combination. This pipeline yields 29k QPC training samples in about 100 hours and saving more than 90%[2] time versus one-by-one check (see Section 3.2).

With sufficient QPC data, the labeling C constitutes a verifiable reward for training a proof RM, enabling an RLVR setup. We adopt a generative RM (Mahan et al., 2024; Zhang et al., 2025a) so that the model can reason step-by-step, generate long thoughts. In the RL stage, however, we observe that combining long reasoning process with binary T/F supervision introduce instability: noisy or speculative thinking process can still reach the correct verdict, thereby assigning incorrect reward to

---

[1] https://artofproblemsolving.com/wiki/index.php/2025_USAMO_Problems

[2] The ratio is approximated by our one-by-one check for test set. See in Appendix A.2.

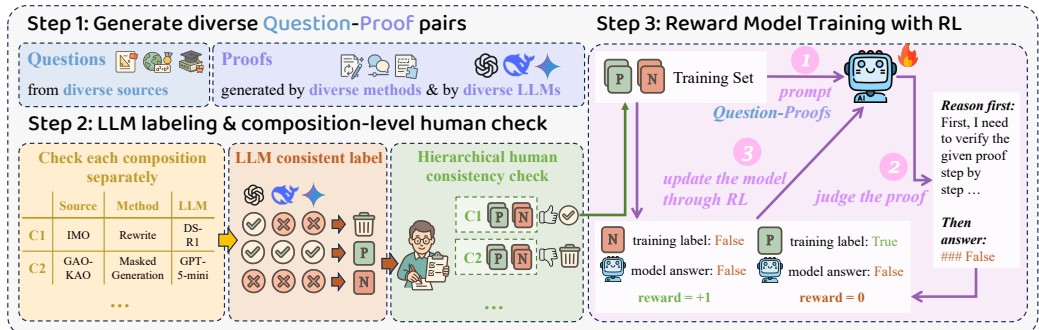

Figure 1: Pipeline of data collection and training of ProofRM.

poor generations and frequently precipitating model collapse. We mitigate it with an "**LLM-as-a-RM for RM**" to supervise the thinking fluency. We also find the drastically varying output lengths strongly correlates to the model collapse, we solve it by a balanced token weight strategy combining sequence-level and token-level token weight. Our modified RL algorithm achieve scalable RL training with increasing performance on more than 312 steps with more than 18k samples and 112M trained tokens. Details can be found in Section 3.3.

Our experiments (Section 4) show that our ProofRM provides accurate evaluations of proofs, outperforming prior baselines and several frontier LLMs. ProofRM also achieves comparable best@k performance—indicating strong utility for test-time scaling, math agents, and data collection.

As a conclusion, our contribution contains:

- Analyzing the asymmetry of verification on math and pointing out the necessary of a proof RM;
- Proposing a question-proof-check collection pipeline to ensure accuracy, scalability and diversity;
- Proposing a practical training recipe to improve training stability and scalability of proof RM RL.

## 2 MOTIVATION: ASYMMETRY OF VERIFICATION OF MATH AND PROOF

### 2.1 PREVIOUS SUCCESS BASED ON STRONG ASYMMETRY

LLMs have recently achieved breakthrough progress on mathematical problem solving. We first highlight that much of this progress can be attributed to a strong **asymmetry in verification difficult** in current math training datasets and benchmarks. In Table 1, we summarize several important math benchmarks, along with the required knowledge of solving them and how to verifying their solutions. A striking pattern is that although solving these problems often requires mathematical knowledge ranging from middle-school level to highly creative advanced techniques, verifying correctness does not become comparably harder, typically can be automatically operated by codes or LMs. Moreover, in terms of reasoning steps, solving these tasks often requires multi-step, exploratory thinking, or branching into alternative solution paths. By contrast, the evaluation only focus on the output. It is precisely this asymmetry in verification that enables large-scale and highly efficient reinforcement learning, which in turn allows models to begin demonstrating strong capabilities on these tasks.

Table 1: A summary of prevailing math benchmarks. Most prevailing math benchmarks explicitly utilize verifiable answer. Num: Numerical answer; Exp: string math expression, Opt: option with candidate answer. *: In NuminaMath, there are about 15% proof data but generally not used in RL.

| Benchmark | Difficulty | Answer Format | Evaluation |
|---|---|---|---|
| GSM8K (Cobbe et al., 2021) | Grade school math knowledge, basic arithmetic operations | Num | string match |
| MATH (Hendrycks et al., 2021) | High school math: intermediate algebra, number theory, ... | Num,Exp | LLM check |
| NuminaMath (LI et al., 2024) | Mainly Chinese high school math exercises, ... | Num,Exp,Opt* | LLM check |
| AIME (Veeraboina, 2024) | Competitive American high school competitions | Num | string match |
| FrontierMath (Glazer et al., 2025) | Experts-crafted, exceptionally challenging modern math | Num,Exp | SymPy match |

Table 2: Methods to make proof-based problems verifiable show vulnerability to shortcut learning.

| Dataset Name | Verified Reward | Shortcut Accuracy | Time Limit | How to guess |
|---|---|---|---|---|
| IneqMath | Bound estimation or relation | 84% | 1min | Assign special values |
| DeepTheorem | Prove or disprove | 70% | 5min | Give counterexamples |
| MathConstruct | Construct part in question | 40% | 15min | Consider simple cases |

## 2.2 LIMITS OF CURRENT ASYMMETRY ON MATH

Although RLVR is currently the most prevalent approach, its limitations are becoming increasingly apparent. First, the process of mathematics is even more important than only the conclusion—as argued by the mathematician William Thurston in his essay: "*Rather, it reflected a continuing desire for human understanding of a proof, in addition to knowledge that the theorem is true.*" (Thurston, 1994). The ideas, tools, lemmas derived to solve the problem matters more and highly develop math field. Yet RLVR provides no supervision of the reasoning process; its effectiveness implicitly rests on the assumption that "*a correct result typically requires a correct derivation*". In contrast, a growing body of evidence shows that RLVR-trained models frequently exhibit correct-answer/incorrect-proof behavior: they may rely on incomplete induction, treat unproven conjectures as facts after checking only a handful of instances, or even invoke nonexistent theorems (Wen et al., 2025b; Tarek & Beheshti, 2025; Petrov et al., 2025).

A more severe problem is that the asymmetry is not from the nature of math but from well-designing of the dataset creator. For more complicated math problem or reserach-level math problem, they are generally proof-based. Non-trivially verifiable results which cannot be easily guessed require significant expert ingenuity and effort. For example, FrontierMath (Glazer et al., 2025) requires 60 expert mathematicians (even one of contributor holds a Fields Medal) to create only 35 research-level problems with a non-trivially verifiable short output. Although this process can create useful benchmark, it is not scalable to construct training data for a reward model training.

## 2.3 SOME ATTEMPT TO VERIFY MATH PROOF

Several work try to provide "a verified answer" for the proof problems. MathConstruct (Balunović et al., 2025) extracts the constructive subproblem in the proof which can be verified by Python code.IneqMath (Sheng et al., 2025) translates the inequality proof into numerical bound estimation and relation prediction ($\geq, \leq, >, <$) tasks. DeepTheorem (Zhang et al., 2025b) translate the groundtruth theorem into statements either entailed by (T) or contradictory from (F) the original theorem and ask about the T/F of these induced statement. These methods indeed enhance the verification asymmetry of proof problems. However, such transformations substantially reduce the difficulty of the original tasks. In practice, these verification of results are highly susceptible to "*shortcut learning*". As illustrated in Table 2, an undergraduate student with modest experience in math competitions can, with little effort, limited time and without constructing a complete proof, achieve a high probability of arriving at the correct answer merely by relying on prior knowledge of the problem's structure. This observation suggests that such transformations are susceptible to hacking and lack appropriate defense mechanisms. Models learn from these supervision could also easily learn unreasonable thinking process, degrading the trustfulness and helpfulness of the models. The detail of the experiments are in Appendix A.3. There are some related direction like formal language proof or MCTS-based PRM, we leave them in Section 5 and discussion in Appendix A.1.

## 2.4 CHECKING THE PROOF ALSO HAS ASYMMETRY OF VERIFICATION

Although verifying full proofs is nontrivial, a **residual verification asymmetry** remains. Constructing a proof entails searching a vast combinatorial space of lemmas and strategies and drawing on tacit mathematical intuition; by contrast, verification largely proceeds linearly along the provided argument and does not require divergent, creative exploration. We therefore believe first training proof RMs expressly for proof verification, and then leveraging these verifiers to improve proof generation via RL, test-time scaling, or data curation/cleaning is a reliable path toward stronger mathematical reasoning.

Table 3: Different input and prompt methods induce distinct proof behaviors.

| Name | Input | Method | Proof Behavior |
|------|-------|--------|----------------|
| Rephrase | QP | Rephrase the proof in the model's own style | An LLM-style proof; mostly correct but may contain step-level flaws |
| Proof | Q | Construct a proof by itself | An LLM-style proof with common, not-too-hard ideas; flaws often include case-analysis mistakes or hallucinations |
| Mask Completion | Q & Masked P | Split the proof, mask key steps, and let the LLM fill them in | A ground-truth-style proof, often with subtle step-level flaws; errors are harder to notice and closer to human mistakes |
| Augment | QP | Change wording or language while preserving content | A ground-truth-style proof retaining original correctness and errors |

## 3 An Scalable and Generalized Proof RM Training Pipeline

In this section we will introduce our RM training pipeline. As mentioned in Section 1, our proof RM should be a generative model, reading the QP pair, generating thinking (checking) process with a final overall check judgment T/F. We first collect QPC dataset with sufficient diversity and quantity, then train an proof RM by RLVR to compare the RM judgement T/F with the labeling C in dataset. The figure 1 provides an illustration of our pipeline.

### 3.1 Automated Diverse Data Collection

Because of the flexibility of proof, a scalable and generalized RM training first demands QP pairs with sufficient diversity. We adapt a multi-dimensional-diversity LLM-aided data generation pipeline (Step 1 in Figure 1). We use three level diversity source: question source, proof generation method, and generating LLM.

**Question-proof source diversity**  Competition-level problems form a core component of our corpus because they are predominantly proof-based and demand challenging reasoning. We draw from OlympiadBench (He et al., 2024) which aggregates math Olympiad problems across difficulty levels and also includes some easier Chinese Gaokao problems. We further complement this with some problems from USAMO and the Putnam competition. The different sources provide diversity in *difficulty level* and *question style*. Most of them have one or more official solution or solution provided by reputable educational institutions, with the latter additionally sampled and verified by the authors. We also collect some authentic student script from partner educational institutions (with proper consent and anonymization handled by the institutions). This student data serves as a valuable source of real-world incorrect proofs or non-perfect correct proofs (like some acceptable leap). Details of data source and student-source data are found in Appendix A.4.1 and A.4.2.

**Proof generation method diversity**  To further enhance the richness and diversity of the dataset, we leverage LLMs to generate additional proofs. We find that using different LLM input and prompt methods can leads to different style of proofs. From the perspective of language style, the spectrum spans from "brief human-like proofs" to "highly authoritative and rigorous official proofs". As for the common error pattern, it varies from "obvious unreasonable idea" to "hard-to-find logic step gap". The theorem space, habitual direction and tools can be also different. Here we summarize these methods in Table 3. A detailed description of these proof-generation strategies and its generated proof can be found in the Appendix A.4.5. This dimension contributes diversity in proof length, level of detail, linguistic style, and error types.

**LLM diversity**  We find different LLMs will generate proof with significantly different length, verbose and style. As a trade-off between cost and diversity, we find combining deepseek-R1, GPT-5-mini (with medium reasoning effort) and Gemini-2.5-flash can provide enough diversity.

We believe there is ample room for further expansion in each of these dimensions, and their free combination will generate highly diverse QP data. Due to resource limitations, we are currently focusing on representative data sources, generation methods, and models. Our future goals include further expanding the exploration space in each dimension once more resources are available.

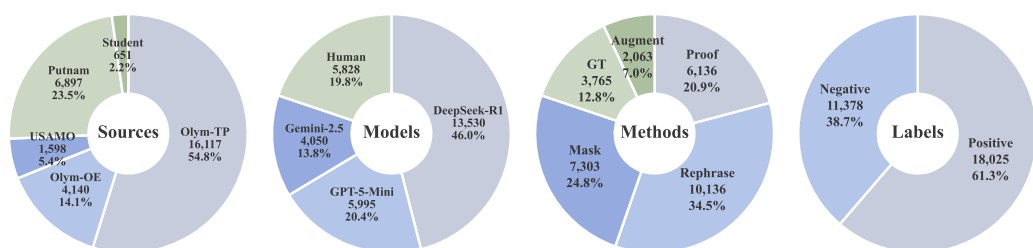

Figure 2: Training data distribution generated by the data pipeline.

We also added some other QP pairs to improve the diversity, including (1) some computational problem where the problem needs a short answer; (2) some artificial degenerated proof (like very short proof, explicit abstain or refusal)

## 3.2 LLM-AIDED CONSISTENT LABELING WITH COMPOSITION-LEVEL HUMAN CHECK

With the collected and generated QP pairs, the next step is to provide an annotation C for each pair. Because the natural language proof cannot be not automatically verified, the expert annotation is the only golden standard, which is too expensive and impossible to be scalable. To balance the scalability and accuracy, we use an LLM-aided consistent labeling with human check. Firstly, we use three LLMs (DeepSeek-R1: 3 times, GPT-5-mini: once, and Gemini-2.5-flash: once—5 times LLM check) to try to check the proof step-by-step and find any flaw in the proof. To get a scalable training set, we only keep the QP pairs that with highly consistency among several times LLM-check and pre-label it by its majority-voting. With the response of our early experiments, we try to improve the LLM-check performance by 1) calling for the reasoning model 2) hinting some common but hard-to-detect error type 3) require a detailed report of error. See the details in Appendix A.4.4.

Then we use a hierarchical human-consistent check where we continuously sample three batches with 5% question-level sampling, with the total QP volume expanded to 0.5%, 1% and 2.5% of QP pairs, and the consistency thresholds were set at 75%, 80% and 90% in each batch. Once, in one composition of question-proof-LLM, the consistency between LLM-check (which has been consistent because of our filter) and human-check is lower than the threshold in one batch, this composition is abandoned. Otherwise, the LLM-check for this composition is considered as reliable. The question with human-checked QP will used as a test set and use the human-annotation as the labeling, and other QP will be used as the training set and use the LLM-annotation as the silver-standard labeling. This approach enables scalable and efficient large-scale data annotation with minimal human effort, while preserving dataset diversity and enforcing a rigorous disjointness between training and test sets.

Our final training set contain 28 combination and 29403 QPC triplets. We show the distribution in the data from the source, prompt methods, LLM and labels dimension in Figure 2. A detailed report of our training data is shown in Table 5 in Appendix A.4.3.

We also need a test set with sufficient diversity to evaluate our proof RM. The first part of our test set is drawn from the 5% sample question in the above human consistency check step. In fact, the sampled 5% questions are further labeled by our human experts one-by-one to provide a golden standard and here we keep both the LLM-consistent and the LLM-inconsistent proofs. These sampled problems are not included in our training corpos.

## 3.3 STABLE RLVR ALGORITHM

We use our collected training data to train our ProofRM. We select our base model as DeepSeek-R1-0528-distill-Qwen3-8B. We find that the base model has strong reasoning ability and can generate reasonable checking process so we do not adopt an additional SFT stage (Appendix A.1). We adopt **verl** (Sheng et al., 2024) as our training framework. We use the GSPO (Zheng et al., 2025) as our base RLVR algorithm where we find it provide more stable training compared to GRPO (Shao et al., 2024) and DAPO (Yu et al., 2025).

However, we find that model collapse will still occur when training after a long stage. We believe it could be due to our binary labeling. Because the supervision is framed as a binary classification, certain early-stage irregular behaviors—such as occasional token repetition, context-irrelevant outputs, reckless guessing, or disproportionately long or short reasoning traces—do not substantially degrade model performance, and may still occasionally lead to correct answers. As a result, positive rewards assigned to these instances can inadvertently reinforce such accidental behaviors, eventually causing irreversible model collapse after a certain stage of training. See some examples in Appendix A.8.

To mitigate this, we introduce an additional mechanism: an LLM-as-a-RM-for-RM. In this setup, the reasoning process and final output generated by the RM are further evaluated by a separate LLM, which we prompt not to delve into the mathematical or logical content, but instead to focus on surface-level fluency and behavioral consistency. Specifically, this auxiliary model checks for signs such as repeated tokens, redundant phrases or logical loops, hasty or unreasoned conclusions, and context-irrelevant token insertions. Whenever such issues are detected, the corresponding sample is labeled as negative regardless of whether the final answer is correct. This procedure prevents frequent model collapse and also serves as a critical diagnostic signal for monitoring whether the supervised training process is proceeding normally. We observe that, since it does not require deep mathematical or logical reasoning, the deepseek-v3.1 model is able to perform this evaluation with high accuracy. In one sampled training interval, between 0.6% and 2% of data instances were flagged as anomalous by the RM-for-RM. Manual inspection of approximately 50 positively flagged cases showed a true positive rate exceeding 96%. Furthermore, in a random sample of 50 unflagged instances, the true negative rate was estimated to be around 96%.

Another issue we discovered for out task is that different **token weighting** strategies led to persistent changes in RM thinking length in the later stages of training. When using a token weighting strategy like DAPO, the output length gradually increased, exhibiting issues of indecision and logical loops. However, when using token weighting strategies like GRPO and GSPO, the model's output length decreased significantly in the later stages of training, exhibiting issues of hasty assertions. We believe that GRPO's inner-sample average $(1/(N|o_i|)$ for the $i$-th sample $o_i$ in $N$ times rollout) can make more significant gradient for shorter response; while the DAPO's inter-sample average $(1/\sum_i |o_i|)$ learn more from longer output. To keep the length stable, we adopt a "*balanced token-weight*" trick, where we set the token weight as a weighted average as $\eta/(N|o_i|) + (1-\eta)/\sum_i |o_i|$, avoiding model's two different unexpected behaviors of the model during long-term training. In our experiments, we find $\eta = 0.6$ can keep the length stable. The ablation result for this trick can be found in Appendix A.10. Other details of our RL recipe are shown in Appendix A.6.

With our modification, our proof RM RL can be stable. It shows an overall trend of continuous growth even after 300 steps of RL training and 18k trained samples. We show our training curve in Figure 3. The test accuracy curve is shown in Appendix A.7.

Figure 3: The training curve shows an increasing trend of reward.

## 4 EXPERIMENTS

### 4.1 GENERALIZED CHECK ACCURACY

Table 4 shows that our ProofRM performs well. The "Mix-up" test set mirrors our diverse training distribution but consists of different questions and is human-labeled item-by-item; the "Student" set holds out student-sourced problems to stress harder cases, labeled by professional coaches. Test sets details are in Appendix A.5. ProofRM beats Distill-Qwen3-8B and matches/exceeds Gemini-2.5-Flash (and slightly GPT-5-mini) on Mix-up, in-

Table 4: Comparison with various baselines on testset

| Model | Mix-up Accuracy | Student Accuracy |
|---|---|---|
| Deepseek V3.1 | 69.55 ±1.79 | 40.61 ±1.69 |
| Gemini-2.5-Flash | 74.52 ±1.66 | 47.41 ±1.75 |
| GPT-5-mini-2025-08-07 | 74.55 ±1.68 | **50.31** ±1.69 |
| Distill-Qwen3-8B | 71.60 ±1.85 | 44.36 ±1.85 |
| Ours | **76.81** ±1.85 | 47.18 ±2.39 |

dicating strong generalization on diverse problem and proof distribution. On Student, it trails some closed-source models but still improves markedly over the base. DeepSeek-V3.1 performs worst in our runs; we observed occasional anomalous outputs (which have been recently reported by many

works) and will re-test after model updates. We also provide some detailed analysis and case study in Appendix A.9.

## 4.2 TEST-TIME SCALING

A good reward model is expected to be able to guide the test-time scaling where the generative candidates should be evaluated and picked (Brown et al., 2024). Following previous works (Frick et al., 2024; Khalifa et al., 2025; Zhao et al., 2025), we use the Best-of-k curve to evaluate the ablity of the RM can guide a more effective test-time scaling. Among all candidate $N$ answers $S_N$, the best-of-$k$ score of a reward model $M : \mathbb{S} \to \{0, 1\}$ with groundtruth $G : \mathbb{S} \to \{0, 1\}$ ($\mathbb{S}$ is the sample space) is

$$\mathbb{E}_{S_k \subseteq S_N} \left[ G(m) \quad \text{where } m = \arg\max\{M(s) | s \in S_k\} \right].$$

$M$ can also be the groundtruth $G$, representing an upper bound of the metric. Because our RM is a generative reward model, we use a 8-times repeated generation and average these 0-1 score to get a continues and robust score $M(s)$.

We generate test set on 18 competition-level math problems from IMO, CMO, USAMO, and other famous competitions from the later 2024 and 2025. We use LLMs including DeepSeek-v3.1, Gemini-2.5-flash, Gemini-2.5-pro, GPT-5, GPT-5-mini and Qwen3-Next-80B-A3B-thinking to generate candidate proofs and we finally collect 48 proofs for each. Because our experiments are designd to show the discrimination of ProofRM, we filter out some too easy or too hard problems where all LLM can (or cannot) provide correct proof. The groundtruth T/F are human-annotated. The Figure 4 shows the best-of-k curve where a large area-under-curve means the better discrimina-

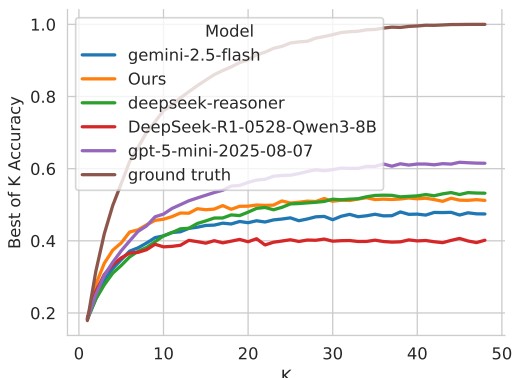

Figure 4: Best-of-k curves for different models.

tion, which can help the test-time scaling and data filtering. Here our models show better performance than our baseline model and even comparable performance with the closed-source strong reasoning models, suggesting good time-time scaling ability.

## 5 RELATED WORK

### 5.1 LLM FOR MATH

An important avenue of LLM reasoning research is math reasoning. Works such as DeepSeek-Math (Shao et al., 2024) and Qwen-Math (Yang et al., 2024) have advanced performance on benchmarks like GSM8K and MATH by strengthening math-oriented pretraining and post-training corpora (including CoT data). Long CoT models typified by o1 (Lea, 2024) have been shown to be particularly effective for math; building on this, DeepSeek-R1 further demonstrates that such long reasoning traces can be automatically induced via RLVR (DeepSeek-AI, 2025). RLVR make large-scale RL-based post-training for math feasible and have become a primary driver of further gains in math capability. Furthermore, test-time scaling methods tailored to math—such as self-reflection (Wang et al., 2023; Yao et al., 2023), parallel thinking (Wen et al., 2025a), math-agent (Xie et al., 2024) or multi-agent systems (Liang et al., 2024; Zhang & Xiong, 2025)—continue to improve LLM performance on math tasks (Wen et al., 2025b). Most strikingly, Google's Gemini reportedly earned a gold medal at the 2025 IMO, marking LLM performance at the level of top human competitors in mathematical contests (Huang & Yang, 2025).

At the same time, verification becomes substantially harder for more frontier math problems, leading to difficulties in data annotation, obtaining verifiable rewards, and the accurate evaluation metrics required for test-time scaling. One line of research tries to reduce verification difficulty by reformatting the problem. A major direction here is about formal-language (FL) proofs, where proofs are written in a formal language (e.g., Lean (Moura & Ullrich, 2021), Isabelle (Wenzel & Berghofer,

2014)) so that symbolic systems can verify them. This approach can efficiently determine proof correctness. Representative resources and benchmarks include MiniF2F (Zheng et al., 2021), ProofNet (Azerbayev et al., 2023), and LeanDojo/ReProver (Yang et al., 2023); Recent efforts in the LLM for FL proof includes DeepSeek-Prover-V2 (Ren et al., 2025), Seed-Prover (Chen et al., 2025b) and so on. However, it faces challenges including data scarcity, a shortage of experts, and inaccuracies or faithfulness issues in translating from natural to formal language. As a result, it is typically viewed not as a substitute for direct natural-language proofs, but as an auxiliary tool for professional mathematicians performing proof checking.

Beyond formal methods, there are works such as Balunović et al. (2025); Sheng et al. (2025) that extract verifiable subproblems, and others such as Glazer et al. (2025) that rely on professional mathematicians to collect data. While these efforts can significantly reduce verification complexity within specific subdomains, they have not yet generalized at scale to broader domains. Recently, several projects have begun to explore training proof capabilities in LLMs directly. Among them, OPC is relatively pioneering: through meticulous human annotation, they curated roughly 5k positive/negative proof examples, providing a valuable dataset for training proof reward models (proof RMs). Nevertheless, the dataset scale is constrained by labor-intensive annotation. Enhancing the scalability and diversity of the labeling pipeline remains necessary, as the primary goal of our work.

### 5.2 REWARD MODEL TRAINING

For most tasks where a verified reward or 0-1 reward is not clearly defined, RL-based post-training needs an RM to evaluate the performance or the alignment degree to human. In RLHF (Ouyang et al., 2022), the RM typically relies on pre-annotated human preference pairs (or rankings) and is trained with a Bradley–Terry objective: via contrastive learning, the RM assigns a higher scalar score to the human-preferred query–response pair. However, purely scalar RMs often underperform on reward tasks that require complex reasoning, and cannot ultilize the test-time scaling. An alternative, simple yet effective approach is LLM-as-a-Judge (Zheng et al., 2023), which directly uses a generative LLM to evaluate the relative quality of a given pair. The original LLM-as-a-Judge is pairwise, making it ill-suited for scenarios such as large-scale data cleaning where pointwise judgments are needed. To address these limitations, some Generative Reward Models (GRMs) fine-tune a generative language model as the RM: the model "thinks" in natural language, expresses the reward as text tokens, and the final numeric reward is extracted from the generated output (e.g., "Ranking: [5]") (Li et al., 2023) or a majority voting to get the output score (Mahan et al., 2024; Zhang et al., 2025a). Building on GRMs, Cloud (Ankner et al., 2024) trains dual heads—a scalar reward head and a language-model head—thereby preserving reasoning and scalable reward to improve training efficiency and reward accuracy. To further enhance cross-task applicability and support multiple evaluation criteria, some RMs introduce meta-criteria training: during inference/training the RM first generates task-specific criteria and then produces a score conditioned on both the criteria and the input (Liu et al., 2025). In this paper, we adopt a GRM to ensure explicit reasoning and choose a base model with strong long-form chain-of-thought capabilities. Since our primary concern is proof correctness, we use the binary (0–1) reward aligning it with standard RLVR practice in math RL and use average score to utilize the test-time scaling. We train the RM in a pointwise style to maximize applicability across downstream use cases (beyond pairwise comparison).

## 6 CONCLUSION

This work revisits the promise and limits of RL with verifiable rewards on math reasoning. We argue that recent gains largely arise from settings with strong verification asymmetry, whereas full proofs lack this asymmetry and demand a reward signal that reads, understands, and checks entire proof. To that end, we introduce a scalable QPC data pipeline that combines real proof sources with LLM-aided multi-dimensional diversity, plus combination-level human auditing to retain only reliable slices at scale. We further present a practical RL recipe for Proof RM—using a generative verifier, an "LLM-as-RM for RM" to supervise thought fluency, and balanced token weighting—that stabilizes training under binary T/F supervision. Empirically, our ProofRM delivers more accurate, generalizable proof judgments than strong baselines and frontier LLMs, and improves best@k utility for test-time scaling.

## 7 FUTURE WORK

We have not been able to fully explore the potential of our method due to limit of resources, in part owing to the degree of complication and lengthiness of our pipeline. For instance, many LLMs, both open-sourced and closed-sourced, remain to be utilized as generation models in our pipeline. Moreover, the available pool of both data sources and prompting strategies can be further expanded upon further experimentation. These aspects serve as the major interest of exploration in the near future. In addition, the current RL scheme still suffers from less-than-ideal training efficiency. We will continue to research on tackling the current bottlenecks and thus enhancing the overall efficiency.

## LLM USAGE

We use LLM to 1) help to search on related work and 2) help to polish the writing of the paper and check the grammar. The authors are always carefully read the response of LLM and use the response of LLM only as a reference and are response for the content.

## ETHICS STATEMENT

Our model is designed exclusively for mathematical proof, and thus does not engage with ethically sensitive tasks or inference about social attributes. Any student-related data have been properly anonymized and de-identified to prevent identity disclosure or re-identification.

## REPRODUCIBILITY STATEMENT

Our paper focus on provide a practical and reproducible proof RL training recipe. To make sure our results are robust and reliable, we report the variance for most of our experiments. We will release most of our training/test data, our training code and test code to ensure the reproducibility later. At the same time, we will release the model parameters of our trained proof RM later.

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

# A APPENDIX

## A.1 DISCUSSION

**About SFT** Because we use LLMs to check each QP pair, these models also generate intermediate reasoning traces. We attempted to use the traces and final answers—restricted to cases with LLM agreement—as SFT data. However, we found that SFT degraded reasoning quality, likely by overfitting to superficial patterns in the data (e.g., stylistic quirks from DeepSeek-R1 or GPT-5-mini). We also observed that, given the strong base model (DeepSeek-R1-0528-distill-Qwen3-8B), the first few RL steps were sufficient for the model to grasp the task and produce outputs in the correct format. Consequently, we adopt a cold-start setting and omit a separate SFT stage.

**About MCTS-based PRM** Outcome Reward Model (ORM) and Process Reward Model (PRM) concern a dimension that is related to—but distinct from—RLVR. RLVR emphasizes verifiability of the final outcome, whereas the ORM–PRM debate centers on the granularity of supervision. In principle, RLVR can be coupled with trajectory sampling (e.g., via MCTS) so that terminal rewards are

back-propagated as expected returns to intermediate steps, thereby providing process supervision. Conversely, an ORM can also perform a holistic review of the entire solution and output a single overall judgment. In this paper, our focus is on issues arising from RLVR rather than adjudicating the ORM vs. PRM debate. That said, given current limitations of PRMs such as step-level label noise and sampling difficulty, we position our model as an ORM with a global, holistic perspective. In fact, the "correct answer with a wrong proof" pathology is not remedied by MCTS-based PRMs: without practically obtainable step-level ground truth, trajectory credit assignment merely redistributes the outcome signal and thus cannot guarantee process correctness.

## A.2 HUMAN CHECK & TEST SET DETAILS

To compare with the naive one-by-one check, we use our best-of-n test set annotation used in Section 4.2 as an approximation. This test set uses 18 problems with 48 answer for each. We deployed a team of four undergraduate students as the annotators, and each answer has been labeled by two annotators. If the results are not consistent between these two annotators, we require the third annotators to make a final decision. The annotator is given a problem and a proof generated by LLM, and need to determine if this proof is valid. Under this process, the annotation of 864 samples took approximately 90 hours of cumulative human labor to complete, translating to a ratio of around 100 hours of human labor per thousand samples. Considering the fact that this process still has much space for optimization, we here use the optimistic estimation of merely 50 hours of annotation per thousand samples, when deploying annotators that excel in math and maximizing the grouping of solutions. Under this approximation, the 24k QPC training samples as forementioned would require a annotation process consisting of at least 1200 hours of human labor. Comparing to the less than 100 hour human annotation required as in our pipeline, this means that our pipeline incorporates a more than 12-fold boost in annotation efficiency.

## A.3 HUMAN SHORTCUT EXPERIMENTS

One of our authors, who has some experience in Math Olympiad but is not a top-level expert, operates these experiments. Here, he first randomly selects 50 problems from IneqMath, 20 problems from DeepTheorem, and 10 problems from MathConstruct, with all the answers to these problems masked. For IneqMath, he utilizes the homogeneity and symmetry of the polynomials to assign special values to the variables, e.g., setting $x = y = z$ to simplify the expression, letting certain variables vanish while allowing the others to approach infinity, or normalizing by assuming $x + y + z = 1$. For DeepTheorem, he tries to give a counterexample and regards the statement as true if and only if he cannot give a counterexample within the time limit. For MathConstruct, without formal proof, he obtains the answer by considering some simple cases, e.g., setting the total number $n = 2$, assuming the function in the statement is a polynomial, or supposing the set that the problem is asking for is an arithmetic sequence. He achieved accuracies of 84%, 70%, and 40% on these three datasets, suggesting that the transformations mentioned in Section 2 greatly reduced the difficulty, thus fundamentally altering the nature of the problems.

## A.4 DATASET DETAILS

### A.4.1 QUESTION-PROBLEM SOURCE

The math part in Olympiad Bench includes International Mathematical Olympiad (IMO), Romanian Master of Mathematics (RMM), American Regions Mathematics League (ARML), Euclid Mathematics Competition (EMC) and European Girls' Mathematical Olympiad (EGMO). Although all of them are math olympiad-level problems, the IMO and RMM are much harder than the others. We also completion the USA Mathematical Olympiad (USAMO) and the William Lowell Putnam Mathematical Competition (Putnam). For all data source, we filter out the geometry problems because from now on the geometry generally use another model to solve instead of an LLM. For the data drawn from Olympiad Bench, we directly use the collected proof in the dataset as the groundtruth. For USAMO, we use the solution provided by a very famous math Olympiad coach Evan Chen in his website https://web.evanchen.cc/problems.html, and we authors sample and check the quality. For Putnam, the Mathematical Association of America (MAA) provides the official solution in https://maa.org/maa-putnam-archive/

### A.4.2 STUDENT-SOURCE DATA

We collect our student-source question-proof pairs from our partner educational institutions. We collect the exercise from some student in a math Olympiad class and the students were informed and consented to the data collection. The papers written by the students were first scanning and anonymized by our partner educational institutions. We use an OCR and human check to make sure the student written problems are faithfully translate to markdown or latex format text. Some empty answers or very short answers have been discarded. All the papers have been scored by two professional Olympiad coaches. If the answer has been scored with full marks by two teachers, we label the proof as T. Otherwise, the proof is F.

The student data contain both non-trivial incorrect proofs and non-perfect correct proofs, which is useful to train a robust and generalized verifier.

---

**Incorrect Proof: Skipping Non-trivial Steps**

**Question.** Given positive integers $a_1, a_2, \ldots, a_{100}$, $b_1, b_2, \ldots, b_{100}$ satisfying $a_1 < a_2 < \cdots < a_{100}, b_1 < b_2 < \cdots < b_{100}$, and such that $a_1 - b_1$, $a_2 - b_2$, $\ldots$, $a_{100} - b_{100}$ are 100 pairwise distinct integers. Find the minimal possible value of $\max\{a_{100}, b_{100}\}$.
**Proof.**
......
Suppose
$C_{i_1} < \cdots < C_{i_1} > C_{i_1+1} > \cdots > C_{i_2} < C_{i_2+1} < \cdots < C_{i_3} > \cdots$
forms an alternating sequence of increasing and decreasing segments.
Thus,
$a_{100} \geq a_1 + 99 + \sum_{2|j}(C_{j+1} - C_j) \geq 100 + \sum_{2|j}(C_{j+1} - C_j)$
$\geq 100 + \max C_i - \min C_i \geq 199,$
and similarly,
$b_{100} \geq b_1 + 99 + \sum_{2|j}(C_{j+1} - C_j) \geq 100 + \sum_{2|j}(C_{j+1} - C_j).$
From the inequality inside the parentheses,
$\geq 100 + \max C_i - \min C_i \geq 199,$
we obtain
$\max\{a_{100}, b_{100}\} \geq 199.$ $\square$

---

We can see that the inequality in red color can't be derived in just a single step, it needs to be nicely proved, not skipped. Students make a lot of these skipping errors in the data.

Below is a fair example of proof which is right but imperfect. The student just used one "Obviously", then the proof is completed and got full mark.

---

**Correct Proof: Using "Commonsense"**

**Question.** Let $a_1, a_2, \ldots, a_8$ to be real numbers. Prove that

$$\sum_{i=1}^{8}(a_i^2 + a_i a_{i+2}) \geq \sum_{i=1}^{8}(a_i a_{i+1} + a_i a_{i+3}) (*)$$

in which i can be seen as modulo 8.
**Proof.**
Obviously, $\sum_{i=1}^{8}(a_i - a_{i+1} + a_{i+2} - a_{i+3})^2 + \sum_{i=1}^{8}(a_i + a_{i+2} - a_{i+3} - a_{i+5})^2 = 0$ (1)
$(1)'sLHS = 8 = ((*)'sLHS - (*)'sRHS)$ $\square$

---

Since both the students and teachers are expert in math olympiads, some non-trivial but frequently used conclusions seems like "commonsense" to them. These "not that non-trivial" proof can train the verifier to balance the degree of skipping details.

### A.4.3 DATA DISTRIBUTION

We list the details of the data distribution in Table 5.

Table 5: The detailed distribution of training data.

| Data Source | Model | Method | Data Num | |
|---|---|---|---|---|
| | | | positive | negative |
| *LLM-aided enhanced (total: 23575)* | | | | |
| OlympiadBench | DeepSeek-R1 | mask_replace | 1210 | 2708 |
| OlympiadBench | DeepSeek-R1 | proof | 1036 | 772 |
| OlympiadBench | DeepSeek-R1 | rephrase | 1385 | 181 |
| OlympiadBench | Gemini 2.5 Flash | proof | 526 | 758 |
| OlympiadBench | Gemini 2.5 Flash | rephrase | 1209 | 339 |
| OlympiadBench | GPT-5-Mini | mask_replace | 589 | 730 |
| OlympiadBench | GPT-5-Mini | proof | 601 | 482 |
| OlympiadBench | GPT-5-Mini | rephrase | 1424 | 136 |
| OlympiadBench-OE | DeepSeek-R1 | rephrase | 1660 | 78 |
| OlympiadBench-OE | DeepSeek-R1 | solution | 0 | 115 |
| Putnam | DeepSeek-R1 | mask_replace | 352 | 785 |
| Putnam | DeepSeek-R1 | proof | 471 | 252 |
| Putnam | DeepSeek-R1 | rephrase | 891 | 207 |
| Putnam | Gemini 2.5 Flash | rephrase | 832 | 386 |
| Putnam | GPT-5-Mini | proof | 466 | 429 |
| Putnam | GPT-5-Mini | rephrase | 944 | 194 |
| USAMO | DeepSeek-R1 | mask_replace | 103 | 826 |
| USAMO | DeepSeek-R1 | proof | 72 | 156 |
| USAMO | DeepSeek-R1 | rephrase | 181 | 89 |
| *Human source data (total: 4339)* | | | | |
| OlympiadBench | - | ground truth | 681 | 0 |
| OlympiadBench-OE | - | ground truth | 1046 | 0 |
| OlympiadBench-CEE | - | ground truth | 1352 | 0 |
| Putnam | - | ground truth | 489 | 0 |
| USAMO | - | ground truth | 120 | 0 |
| Student | - | ground truth | 55 | 22 |
| Student | Deepseek-V3.1 | augment | 165 | 122 |
| Student | Deepseek-V3.1 | translate | 165 | 122 |
| *Auxiliary data (total: 1489)* | | | | |
| - | - | naive negative | 0 | 1489 |

### A.4.4 LLM CHECK

Here we first provide our LLM system prompt of our LLM check. The principle of the prompt is:

- The four types of error guide LLM to think comprehensively and try to find different type of error.

- Detailed instruction about *"what kind of proof could be considered as correct"*.

- Some hint about common error type like *Special-case testing* or *non-standard symbols*.

---

**Prompt of LLM check**

You are an expert math Olympiad proof verifier. Given a problem and its proposed proof, rigorously verify the proof's correctness by evaluating:
(1) **Logical Soundness**
- Every inference must be valid with clear justification. There is no logical fallacies.
- All lemmas/theorems must have satisfied conditions.
- No critical gaps in reasoning (minor omissions acceptable).
- *Explicit ban:* Special-case testing != valid proof
(2) **Computational Accuracy**
- All calculations error-free
- No unjustified approximations: Numerical estimates should generally not be used unless directly proving magnitude relationships between quantities.
(3) **Structural Completeness**
- Full coverage of problem required statement: if there is a required value or expression, the value or expression has been explicitly answered and correct.
- If a specific final answer is required, the result should be fully simplified. If an equivalent further simpler form exists, mark the proof incorrect (because it is incomplete) (e.g.,if closed form exists, a recurrence is not enough)
- Well-organized proof flow.
(4) **Notational Rigor**
- Correct mathematical notation
- All definitions/theorems explicitly stated
- No ambiguous or non-standard symbols
**Output Format:**
```json
"condition1_satisfied": true/false,
"condition2_satisfied": true/false,
"condition3_satisfied": true/false,
"condition4_satisfied": true/false,
"proof_correct": true/false // Only true if ALL conditions satisfied
```

---

### A.4.5 GENERATED NEGATIVE SAMPLE ANALYSIS

We observe that our pipeline is able to generate a great variety of negative samples, with varying degree of obviousness and imitation of human behaviors. Hereby we provide a detailed example analysis on some most commonly seen negative samples genres generated by our pipeline. We first present some classical mistakes often observed in LLM generated solutions of mathematical problems. One of which is only proving one side of a statement that requires both sufficiency and necessity, leaving the other side with few strokes of reasoning or some easy case analysis. The latter confusion between simple case analysis and full proof is also a mistake among the most commonly seen ones in our generated mathematical solutions.

---

**Negative sample with only one side of reasoning**

**Question.** A repunit is a positive integer whose digits in base 10 are all ones. Find all polynomials $f$ with real coefficients such that if $n$ is a repunit, then so is $f(n)$.

---

**Answer 1.** The polynomials that satisfy the condition are those of the form

$$f(x) = \frac{10^j (9x+1)^d - 1}{9}$$

for integers $d \geq 0$ and $j \geq 1 - d$.

To verify,

......

All such polynomials have real coefficients and map repunits to repunits, and there are no others due to the requirement that the functional equation holds for all repunits, which are infinite and distinct.

$$\boxed{f(x) = \frac{10^j \cdot (9x+1)^d - 1}{9} \quad \text{for integers} \quad d \geq 0 \quad \text{and} \quad j \geq 1 - d}$$

**Answer 2.**

......

Conversely, every polynomial satisfying the condition must be of this form. Constant solutions arise when $d = 0$, leading to $f(x) = \frac{10^M - 1}{9} = R_M$ for $M \geq 1$, which are constant polynomials taking repunit values. For $d \geq 1$, non-constant polynomials are obtained. The following cases exemplify the form: - $d = 0$, $M \geq 1$: Constants $f(x) = R_M$. - $d = 1$, $M = 0$: Identity $f(x) = x$. - $d = 1$, $M = 1$: $f(x) = 10x + 1$, mapping $R_k$ to $R_{k+1}$. - $d = 2$, $M = -1$: $f(x) = \frac{(9x+1)^2/10-1}{9} = \frac{81x^2+18x-9}{90}$, mapping $R_k$ to $R_{2k-1}$. - $d = 2$, $M = 0$: $f(x) = \frac{1 \cdot (9x+1)^2 - 1}{9} = 9x^2 + 2x$, mapping $R_k$ to $R_{2k}$.

All polynomials meeting the condition are included, with the parameters $d$ and $M$ covering both constant and non-constant cases. Thus, the solution set is characterized by the given form.

$$\boxed{f(x) = \frac{10^M (9x+1)^d - 1}{9} \quad \text{for integers} \quad d \geq 0 \quad \text{and} \quad M \geq 1 - d}$$

Another classical negative sample genre involves the hallucination of LLMs, where the generated solution contains some not readily available, if existent at all, mathematical results. One needs to point out that such hallucinated results are not dedicated to LLMs, as human might also make up some non-existing results. However, in the case of human, the act of making up is usually deliberate, often a crude attempt to trick the reader.

**Negative sample with not readily available results**

**Question 1.** Let $a, b, c, d$ be positive real numbers such that

$$abcd = 1 \quad \text{and} \quad a + b + c + d > \frac{a}{b} + \frac{b}{c} + \frac{c}{d} + \frac{d}{a}$$

Prove that

$$a + b + c + d < \frac{b}{a} + \frac{c}{b} + \frac{d}{c} + \frac{a}{d}$$

**Answer 1.**

. . .

A known inequality states that for any positive real numbers $x_1, x_2, x_3, x_4$,

$$\frac{x_1}{x_2} + \frac{x_2}{x_3} + \frac{x_3}{x_4} + \frac{x_4}{x_1} \geq \frac{x_1 + x_2 + x_3 + x_4}{\sqrt[4]{x_1 x_2 x_3 x_4}}.$$

Applying this with $(x_1, x_2, x_3, x_4) = (a, b, c, d)$ and using $abcd = 1$,

. . .

**Question 2.** Let $a, b, c, d$ be positive integers such that $ad \neq bc$ and $\gcd(a, b, c, d) = 1$. Prove that, as $n$ runs through the positive integers, the values $\gcd(an + b, cn + d)$ may achieve form the set of all positive divisors of some integer.

**Answer 2.**

...

A theorem by E.J. Cohn states: Let $P(x) = ax + b$ and $Q(x) = cx + d$ be linear polynomials with integer coefficients. The set of values $\{\gcd(P(n), Q(n)) \mid n \in \mathbb{Z}\}$ is precisely the set of all positive divisors of the integer $M = \gcd(|ad - bc|, \gcd(a, b) \gcd(c, d))$.

...

The examples listed above are all generated with the *proof* method, as is previously mentioned in our pipeline. Indeed, for these common negative sample genres, the *proof* method is more likely to generate them. As we progress towards more diverse and less naive negative samples, the negative samples generated with the *mask_replace* method takes up a dominant position. As a rule of thumb, these negative samples often involve overlooking certain preconditions or taking conditions for granted. Among these, one example is to ignore the requirements for theorems and mathematical operations. In the following example, the solution, generated by *proof*, overlooks the requirement of continuity for taking the limit.

**Negative sample overlooking preconditions 1**

**Question.** Does there exist a pair $(g, h)$ of functions $g, h : \mathbb{R} \to \mathbb{R}$ such that the only function $f : \mathbb{R} \to \mathbb{R}$ satisfying $f(g(x)) = g(f(x))$ and $f(h(x)) = h(f(x))$ for all $x \in \mathbb{R}$ is the identity function $f(x) \equiv x$ ?

**Answer.**

...

Step 4: Show that $f(x) = x$ for all $x \geq 0$. Let $x \geq 0$. Define $F(x) = f(x) - x$.

...

Let's prove $F(x) = 0$ for all $x \geq 0$. Suppose there exists some $x_0 \geq 0$ such that $F(x_0) \neq 0$. Since $F(n) = 0$ for $n \in \mathbb{Z}$, $x_0$ must not be an integer. Since $F(x)$ is periodic with period 1, we can consider $x_0 \in (0, 1)$. Case A: $F(x_0) > 0$. Then $f(x_0) > x_0$. From $F(x_0^2) = 2x_0 F(x_0) + (F(x_0))^2$, since $x_0 \in (0, 1)$ and $F(x_0) > 0$, we have $2x_0 > 0$ and $(F(x_0))^2 > 0$. Thus $F(x_0^2) > 2x_0 F(x_0)$. Let $x_k = x_0^{2^k}$ for $k \in \mathbb{N}_0$. Then $x_k \in (0, 1)$ and $x_k \to 0$ as $k \to \infty$. Also, $F(x_k) \to F(0) = 0$ as $k \to \infty$.

...

A more obscure example of overlooking preconditions is contributed by the *mask_replace* method, with a combinatorics problem as follows. In this example, the generated solution overlooked the part on the construction of so called "traps", making the entire logic chain of the solution invalid. These kind of mistakes are also commonly seen in human-written solutions, either due to negligence or falsely beliefs that such preconditions are trivial.

**Negative sample overlooking preconditions 2**

**Question.** The Y2K Game is played on a $1 \times 2000$ grid as follows.Two players in turn write either an S or an O in an empty square.The first player who produces three consecutive boxes that spell SOS wins.If all boxes are filled without producing SOS then the game is a draw.Prove that the second player has a winning strategy.

**Answer.**

The main insight is that a construct of the form

$$S \,\square\,\square\, S$$

(here the $\square$ is blank) will kill any player which plays inside it. We call this a trap accordingly. The specific strategy is as follows: The core strategy of the second player is to win by setting traps. A trap is a structure like $S \,\square\,\square\, S$. If a player is forced to place a piece in either of the two empty squares, the opponent can immediately complete an SOS and win. On her turn, the second player prioritizes attempting to construct a trap: she selects an existing S and places another S on an empty square three positions away (thus creating two S's with two empty squares between them). However, this must meet three conditions: (1) The two squares between the two S's are both empty; (2) Placing an S in this position

does not immediately form an SOS; (3) This move will not cause her to lose immediately. If she cannot construct a trap, she will adopt the safe move method described in subsequent propositions. Since the board has 2000 squares, she can eventually create a trap. Once a trap is formed, when the first player is forced to move within the trap (as all squares will eventually be filled), the second player only needs to place the appropriate letter in the remaining empty square (if the opponent places an O in the first empty square, she places an S to form SOS; if the opponent places an S, she places an O to also form SOS) to win.

**Claim.** The second player always has a move which prevents her from losing.

**proof.** Since there are an odd number of empty squares at the start of the second player's turn, there must be a square which is bordered by either two filled or two empty squares. The second player can then play $O$ in this square, which is always safe.

Together these two claims finish the problem.

Additionally, another genre of negative samples generated by our pipeline that aligns with common human mistakes is the misuse of "Without Loss of Generality" (WLOG). This type of mistake is ever-present in all levels of mathematical problem solving. The following is an example of such negative samples, also generated by the *mask_replace* method. The solution automatically converted the general polygon case to the quadrilateral case, which is neither explicitly stated nor obvious. The rationale for such a conversion can be obtained from the explanation excerpted from the reference solution. Without such reasoning, one is unable to distinguish between a valid solution and a mere analysis of a easy case.

---

**Negative sample with misuse of WLOG**

**Question.** A convex polygon $\mathcal{P}$ in the plane is dissected into smaller convex polygons by drawing all of its diagonals.The lengths of all sides and all diagonals of the polygon $\mathcal{P}$ are rational numbers. Prove that the lengths of all sides of all polygons in the dissection are also rational numbers.

**Answer.**
Since we know that the side lengths $a$, $b$, $c$, $AD^2$, $BD^2$, and $CD^2$ are all rational numbers, we can set up a linear system of equations in terms of $x$, $y$, and $z$, and use the relation $x + y + z = 1$ to solve it. First, subtracting equation (2) from equation (1) gives:

$$\left(a^2 z + c^2 x\right) - \left(b^2 z + c^2 y\right) = (BD^2 + W) - (AD^2 + W) \implies c^2(x-y) + (a^2 - b^2)z = BD^2 - AD^2,$$

which results in a rational number. Similarly, subtracting equation (3) from equation (2) gives:

$$\left(a^2 y + b^2 x\right) - \left(a^2 z + c^2 x\right) = (CD^2 + W) - (BD^2 + W) \implies a^2(y-z) + (b^2 - c^2)x = CD^2 - BD^2,$$

...

**Reference Solution.** Suppose $AB$ is a side of a polygon in the dissection, lying on diagonal $XY$, with $X$, $A$, $B$, $Y$ in that order. Then

$$AB = XY - XA - YB.$$

In this way, we see that it actually just suffices to prove the result for a quadrilateral.

...

---

A further illustration of the negative samples is characterized by the inability to fully answer the question. Commonly observed with *mask_replace* generated negative samples, the solution often, possibly owing to overlooking certain parts of the question or simply unable to solve the missing part, fails to address all aspects of the problem correctly. This circumstance often occurs when multiple statements are required by the problem. The following example demonstrates one such scenario. The example, despite completing the majority of the proof, failed to establish the final closed-form expression for the number of ways to play the cards.

---

**Negative sample with incomplete solution**

**Question.** A game of solitaire is played with $R$ red cards, $W$ white cards, and $B$ blue cards. A player plays all the cards one at a time. With each play he accumulates a penalty. If he plays a blue card, then he is charged a penalty which is the number of white cards still in his hand. If he plays a white card, then he is charged a penalty which is twice the number of

---

red cards still in his hand. If he plays a red card, then he is charged a penalty which is three times the number of blue cards still in his hand.

Find, as a function of $R$, $W$, and $B$, the minimal total penalty a player can amass and the number of ways in which this minimum can be achieved.

**Answer.** The minimum penalty is

$$f(B, W, R) = \min(BW, 2WR, 3RB)$$

or equivalently, the natural guess of "discard all cards of one color first" is actually optimal (though not necessarily unique).

This can be proven directly by induction. Indeed the base case $BWR = 0$ (in which case zero penalty is clearly achievable).

The inductive step follows from

$$f(B, W, R) = \min \begin{cases} f(B - 1, W, R) + W \\ f(B, W - 1, R) + 2R \\ f(B, W, R - 1) + 3B. \end{cases}$$

It remains to characterize the strategies. This is an annoying calculation, so we just state the result.

Let the minimum penalty be $M = \min(BW, 2WR, 3RB)$. Then the number of optimal coloring sequences $h(B, W, R)$ is defined by the recurrence: $h(0, 0, 0) = 1$, and when $B + W + R > 0$,

$$h(B, W, R) = \sum_{\substack{c \in \{\text{blue,white,red}\} \\ \text{if } c=\text{blue, then } BW=M; \\ \text{if } c=\text{white, then } 2WR=M; \\ \text{if } c=\text{red, then } 3RB=M}} h(\text{state after removing } c)$$

where blue, white, and red correspond to the states after removing one card of that color: $(B - 1, W, R)$, $(B, W - 1, R)$, and $(B, W, R - 1)$, respectively.

Thus, when the cards are distinguishable, the total number of schemes is $h(B, W, R) \times (B!\, W!\, R!)$.

Finally, for method *rephrase*, most of the time it generates positive samples, but occasionally it also produces negative samples. The negative samples generated by *rephrase* often contain subtle mistakes, hidden inside large chunks of correct reasoning, making it harder to spot. The following is an example of such negative samples, a variant of the previous sample type, nevertheless with a much longer solution.

**Negative sample with subtle mistakes**

**Question.** Given six positive numbers $a, b, c, d, e, f$ such that $a < b < c < d < e < f$. Let $a + c + e = S$ and $b + d + f = T$. Prove that

$$2ST > \sqrt{3(S + T)(S(bd + bf + df) + T(ac + ae + ce))}.$$

**Answer.**

...¡1.5k words establishing the inequality¿ ...

This is precisely the Arithmetic Mean-Geometric Mean (AM-GM) inequality. Since $a, b, f$ are positive numbers, $A_1 = 3b(a + 4b + f)$ is positive, and $B_1 = (a + 2b)(b + 2f) + (2b + f)(2a + b)$ is positive. The AM-GM inequality states that for positive numbers $A_1, B_1$, $\frac{A_1 + B_1}{2} \geq \sqrt{A_1 B_1}$. Equality holds if and only if $A_1 = B_1$. Let's check if $A_1 = B_1$ in our case: $3ab + 12b^2 + 3bf = 5ab + 4af + 4b^2 + 5bf$. This simplifies to $8b^2 - 2ab - 2bf - 4af = 0$. In general, this equality does not hold. For instance, if $a = 1, b = 2, f = 3$: $A_1 = 3(2)(1 + 4(2) + 3) = 6(12) = 72$. $B_1 = (1 + 4)(2 + 6) + (2(1) + 2)(2(2) + 3) = 5(8) + (4)(7) = 40 + 28 = 68$. Since $A_1 \neq B_1$ for these values, the inequality is strict. In fact, $A_1$ and $B_1$ are generally different for $a, b, f > 0$. Therefore, $\frac{A_1 + B_1}{2} > \sqrt{A_1 B_1}$ holds as a strict inequality.

> Since the inequality holds for the transformed variables, and we ensured that our transformations strictly increased the right-hand side while keeping the left-hand side fixed, the original strict inequality must also hold.
> The proof is complete.

### A.4.6 DETAILS OF HUMAN CHECK

The Table 6 shows the details of our human check.

Table 6: Details of human check results. TP and TN stands for True Positive and True Negative respectively, equivalently cases where the results of LLM check align with human check results.

| Dataset | Model | Method | Positive | | Negative | | Consistency | | Samples | |
|---|---|---|---|---|---|---|---|---|---|---|
| | | | Amount | TP ratio | Amount | TN ratio | Ratio | Accepted | Checked | Generated |
| Olympiad-Bench | Deepseek-R1-0528 | mask | 11 | 0.82 | 4 | 0.75 | 0.8 | False | 26 | 105 |
| Olympiad-Bench | Deepseek-R1-0528 | proof | 9 | 0.89 | 16 | 0.94 | 0.92 | True | 30 | 72 |
| Olympiad-Bench | Deepseek-R1-0528 | rephrase | 10 | 1 | 10 | 0.9 | 0.95 | True | 30 | 111 |
| Olympiad-Bench | Deepseek-V3.1 | mask | 4 | 0.75 | 14 | 0.43 | 0.5 | False | 26 | 104 |
| Olympiad-Bench | Deepseek-V3.1 | proof | 11 | 1 | 16 | 0.88 | 0.93 | True | 36 | 72 |
| Olympiad-Bench | Gemini-2.5-flash | mask | 16 | 0.88 | 3 | 0.33 | 0.79 | False | 30 | 96 |
| Olympiad-Bench | Gemini-2.5-flash | proof | 16 | 0.81 | 8 | 1 | 0.87 | False | 30 | 72 |
| Olympiad-Bench | Gemini-2.5-flash | rephrase | 24 | 0.83 | 1 | 1 | 0.84 | False | 30 | 111 |
| Olympiad-Bench | GPT-5-mini | mask | 10 | 1 | 12 | 0.67 | 0.82 | False | 30 | 102 |
| Olympiad-Bench | GPT-5-mini | proof | 10 | 1 | 17 | 1 | 1 | True | 30 | 72 |
| Olympiad-Bench | GPT-5-mini | rephrase | 21 | 1 | 1 | 0 | 0.95 | True | 30 | 111 |
| Putnam | Deepseek-R1-0528 | mask | 5 | 1 | 11 | 0.91 | 0.94 | True | 22 | 69 |
| Putnam | Deepseek-R1-0528 | proof | 9 | 1 | 8 | 1 | 1 | True | 30 | 51 |
| Putnam | Deepseek-R1-0528 | rephrase | 18 | 1 | 5 | 0.8 | 0.96 | True | 30 | 78 |
| Putnam | Deepseek-V3.1 | mask | 7 | 1 | 15 | 0.93 | 0.95 | True | 25 | 55 |
| Putnam | Gemini-2.5-flash | mask | 12 | 0.75 | 10 | 0.6 | 0.68 | False | 29 | 69 |
| Putnam | Gemini-2.5-flash | proof | 19 | 0.68 | 5 | 0.4 | 0.62 | False | 30 | 51 |
| Putnam | Gemini-2.5-flash | rephrase | 19 | 0.95 | 4 | 0.75 | 0.92 | True | 30 | 78 |
| Putnam | GPT-5-mini | mask | 16 | 0.94 | 4 | 1 | 0.95 | True | 30 | 63 |
| Putnam | GPT-5-mini | proof | 16 | 0.88 | 7 | 1 | 0.92 | True | 30 | 51 |
| Putnam | GPT-5-mini | rephrase | 20 | 0.95 | 3 | 0.67 | 0.91 | True | 30 | 78 |
| Usamo | Deepseek-R1-0528 | mask | 11 | 0.82 | 11 | 1 | 0.91 | True | 30 | 61 |
| Usamo | Deepseek-R1-0528 | proof | 7 | 0.86 | 13 | 1 | 0.95 | True | 27 | 30 |
| Usamo | Deepseek-R1-0528 | rephrase | 11 | 0.91 | 4 | 1 | 0.93 | True | 25 | 33 |
| Usamo | Deepseek-V3.1 | mask | 1 | 1 | 16 | 0.69 | 0.71 | False | 22 | 27 |
| Usamo | Deepseek-V3.1 | proof | 8 | 1 | 15 | 0.93 | 0.95 | True | 26 | 30 |
| Usamo | Deepseek-V3.1 | rephrase | 12 | 0.83 | 13 | 0.69 | 0.76 | False | 30 | 33 |
| Usamo | GPT-5-mini | mask | 6 | 1 | 17 | 0.47 | 0.61 | False | 27 | 30 |
| Usamo | GPT-5-mini | rephrase | 14 | 1 | 13 | 0.31 | 0.67 | False | 30 | 30 |
| Usamo | GPT-5-mini | proof | 5 | 1 | 23 | 0.91 | 0.93 | True | 30 | 30 |

### A.5 TEST SET DETAILS

We have three parts of the test sets, named as:

- **Mixed-up**: randomly split from our training set in the question level, but keep not only the QAs with consistent LLM judgment but also the ones with inconsistent judgment. The test set share a similar distribution of the problem source, extended methods and extending LLM but include different questions (as well as their proofs) to avoid data leakage. This test set mainly reflects the general performance improvement on diverse QA pairs as well as an in-domain performance improvement introduced by our Proof RM RL. This set contains 238 QA pairs from 51 questions, where 141 samples among them are positive and 97 samples are negative.

- **Student-sourced**: This test set is split from the collected student-sourced data and directly use the couch score as the label. We first remove these trivial negative samples like empty answers or very short answer. The test set contains 204 QA pairs from 19 questions, where 139 samples are positive and 65 samples are negative. This set reflects the performance on the "harder" data samples, where the difficulty lies in proofs that are more refined and concise, allowing for occasional justified leaps (e.g., *by similar reasoning*), more errors and logical flaws that are harder to detect and sometimes deliberately concealed, and exhibiting a linguistic style and generation habits that differ markedly from those of large language models' own proofs. At the same time, this test set reflects the model's "*extensibility*": this portion of data is substantially different from the model's priors, whether those priors

derive from official answers seen in pretraining or from LLM-generated proofs observed through distillation of larger models.

- **Best-of-k test**: To test whether our RM is helpful to guide Math Olympic solver in test-time scaling setting, we additionally collect new problems in 2024 and 2025 with LLM-generated proofs for each problem. These LLMs includes DeepSeek-v3.1, Gemini-2.5-flash, Gemini-2.5-pro, GPT-5, GPT-5-mini and Qwen3-Next-80B-A3B-thinking. We use different LLMs to provide sufficiently diverse proof and simulate the situation where the performance of the proofs are various. We also manually filtered some too easy or too hard problems, or some too short (naive negative) or truncated proofs. We finally collect 18 problems and 48 proofs for each problems.

## A.6 RL HYPER-PARAMETERS

We list the RL hyperparameters in Table 7. In our RL scheme, we adopted the **verl** framework Sheng et al. (2024).

| Hyperparameters | | |
|---|---|---|
| **optimizer** | learning rate | 5e-7 |
| | batch size | 480 |
| | scheduler | cosine |
| | weight decay | 0.1 |
| | warm up ratio | 0.05 |
| **rollout** | rollout num | 8 |
| | temperature | 0.6 |
| | top_p | 0.9 |
| **actor** | clip ratio low | 3e-4 |
| | clip ratio high | 4e-4 |
| | use kl loss | True |
| | kl loss coef | 0.05 |
| **data** | max prompt length | 8,192 |
| | max response length | 16,000 |

Table 7: RL hyperparameters. We adopted the one-step-delay training method provided by VeRL to avoid the latency of invoking LLM-as-an-RM for the reward model.

## A.7 TEST ACCURACY CURVE

Here we test different checkpoint in our training stage, and draw the curve of test accuracy against trained samples in Figure 5. The curve also shows a similar increasing trend with the training curve shown in Figure 3.

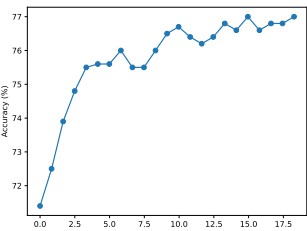

Figure 5: Accuracy on the testset.

## A.8 MODEL COLLAPSE AND LLM-AS-AN-RM-FOR-RM

In our RL training stage, model collapse occasional occurs where the accuracy suddenly drops and at the same time the output length also shows significant variant. We find just before the accuracy finally drop, there have some "aura" where the thinking process of the model has been seemed odd but it finally give a correct answer, as the example in the following.

---

**Bad Case: verbose and repeat**

......
Since this is a text, I need to output at the end whether the proof is correct or not, but in the output, it says to output True or False, but in the example, it says ### True or False, but in the user message, it says "### True or False" but in the example, it says "### True" but in the user message, it says "### True or False" but in the initial, it says "### True or False" but in the example, it says "### True" but I think it's to output whether the proof is correct or not.

......
Let me find errors.
First, in the proof, in step 4, it claims k=2-sqrt(3), but is it correct?
But I need to verify the proof.
Now, let me find errors in the proof.
First, in the proof, in step 1, angle bisector, etc.
But let me find one error.
In the proof, in step 2, coordinate geometry, etc.
In the proof, in step 2, it says "using coordinates, E has x-coordinate 2ac/(a+c), D has d=2a^2/(a+c), etc."
But in the proof, it's messy.

......
In proof, in step 4, it has incorrect derivation.
Therefore, there is error in proof.
The proof contains multiple errors, including logical inconsistencies, incorrect calculations, and invalid claims. Key errors include incorrect derivation of k, invalid assumptions, and computational mistakes. Therefore, the proof is incorrect.
**### False**

---

We can easily find some repeated and nonsense sentences (red parts), but finally the model provide a correct verdict (green parts), which means the reward will be positive and the repeated tokens will be further learned by models. Just after the aura appear with several steps, the model has finally collapse and it cannot give a final answer. Try to solve the problem, we need to find and rectify it once the aura appears.

Because this odd behavior is somehow easy to find, we just adopt a pretrained LLM to read the RM's output. We call the model as RM-for-RM. Here, we provide only the generation (Check) of RM to the RM-for-RM instead of the input-generation pairs (QPC) to avoid the RM-for-RM Distracted by the math problem. Once the RM-for-RM find some odd behavior in the generation, whatever the T/F prediction is correct, the check will be set as incorrect check and the reward will be set to 0. This rectification can improve the training stability. We also use it as a more sensitive metric to avoid model collapse. If the odd generation ratio is with continuous five steps, we will stop the training and resume from the nearest checkpoint. here.

**Why should surface-level fluency checks prevent collapse?** The collapse we observed in our training is usually due to some incorrectly encourage introduced by these "surface-level" fault. We observe that in some steps, some checking generation will include some repeated phrases or words (like "so so so so") but finally leads to a correct T/F judgment. If our RL process incorrectly encourage these samples, after several steps, the model may learn to generate repeated words more and more frequently, which finally lead to the collapse. So it is important to find these unexpected behaviors as early as possible and avoid encouraging them. In our training stage, we find at some

steps, the odd generation occurred, and after 5-10 steps the entropy will significantly explode and the reward will drops to almost 0.

### A.9 ERROR TYPE BREAKDOWN AND CASE STUDY OF PROOFRM

In this section, we provide some detailed analysis of our behavior ProofRM. First, we find that our model has been successfully learn to recognize some deduction error in the proof, which is hard to recognize for the untrained baseline. Below we show some examples:

---

**Example of learned deduction errors**

**Question.** Let $h$ and $k$ be positive integers. Prove that for every $\epsilon > 0$, there are positive integers $m$ and $n$ such that
$$\epsilon < |h\sqrt{m} - k\sqrt{n}| < 2\epsilon.$$

**Checked Proof.**

......
**Case 1:** $r = \left(\frac{k}{h}\right)^2$ **is an integer.**
...(establishing the following claim)...
For $\epsilon > 0$, we need to choose $n$ such that

$$\epsilon < \frac{h^2}{2k\sqrt{n} + h} \quad \text{and} \quad \frac{h^2}{k\sqrt{n}} < 2\epsilon.$$

The inequality $\frac{h^2}{k\sqrt{n}} < 2\epsilon$ implies $\sqrt{n} > \frac{h^2}{2k\epsilon}$. The inequality $\epsilon < \frac{h^2}{2k\sqrt{n}+h}$ is equivalent to $\epsilon(2k\sqrt{n}+h) < h^2$, so $2k\epsilon\sqrt{n} + h\epsilon < h^2$, which gives $2k\epsilon\sqrt{n} < h^2 - h\epsilon$. For $\epsilon < h$ (which holds for sufficiently small $\epsilon$, and for large $\epsilon$ we adjust as below), we have $h^2 - h\epsilon > 0$, so

$$\sqrt{n} < \frac{h^2 - h\epsilon}{2k\epsilon}.$$

Thus,

$$\frac{h^2}{2k\epsilon} < \sqrt{n} < \frac{h^2 - h\epsilon}{2k\epsilon}.$$

The interval $\left(\frac{h^2}{2k\epsilon}, \frac{h^2 - h\epsilon}{2k\epsilon}\right)$ is non-empty when $\frac{h^2 - h\epsilon}{2k\epsilon} > \frac{h^2}{2k\epsilon}$, which simplifies to $-h\epsilon > 0$, a contradiction. However, the actual range for $\sqrt{n}$ derives from the bounds:

$$|h\sqrt{m} - k\sqrt{n}| = \frac{h}{\sqrt{c^2 n + 1} + c\sqrt{n}} > \frac{h}{2c\sqrt{n} + 1} = \frac{h^2}{2k\sqrt{n} + h},$$

and

$$|h\sqrt{m} - k\sqrt{n}| < \frac{h}{c\sqrt{n}} = \frac{h^2}{k\sqrt{n}}.$$

As $n$ increases, the expression $\frac{h}{\sqrt{c^2 n + 1} + c\sqrt{n}}$ decreases continuously to 0. Therefore, for any $\epsilon > 0$, since the expression takes all values in some interval $(0, M]$ for $M = |h\sqrt{c^2 \cdot 1 + 1} - k\sqrt{1}| > 0$ (e.g., for $n = 1$), and decreases, there exists a smallest $n$ such that $|h\sqrt{m} - k\sqrt{n}| \leq 2\epsilon$ (for large $n$, it is small). For this $n$, if $|h\sqrt{m} - k\sqrt{n}| > \epsilon$, we are done. If not, then at the next smaller $n$, the value is larger, but in all computations, for the chosen $n$ in the sequence, it satisfies both inequalities strictly as the sequence strictly decreases and the inequalities are open.
For large $\epsilon$, since $|h\sqrt{m} - k\sqrt{n}|$ can be made arbitrarily large (e.g., fix $n = 1$, let $m \to \infty$), and the set of values has no upper bound, and the expression with $m = c^2 n + 1$ decreases to 0, the values cover an interval, so we can find $n$ such that $\epsilon < |h\sqrt{m} - k\sqrt{n}| < 2\epsilon$.
**Case 2:** $r$ **is not an integer.**
Write $r = \frac{a}{b}$ in lowest terms with $\gcd(a, b) = 1$, so $a$ and $b$ are positive integers with $b \geq 2$.
The sequence $rn = \frac{a^2}{b^2} n$ modulo 1 takes values $\frac{j}{b^2}$ for $j = 0, 1, \ldots, b^2 - 1$. The distance to the nearest integer, $d(n) = \min_v |v - rn|$, satisfies $\max_n d(n) = D > 0$, where $D$ depends

---

on $b$. Specifically, $D = \max_j \min\left(\frac{j}{b^2}, 1 - \frac{j}{b^2}\right)$. For example, with $b = 2$, $D = 0.5$; with $b = 3$, $D \approx 0.444$. Choose an $n$ such that $d(n) = D$, which exists by periodicity modulo $b^2$. For this $n$, choose $m$ as the integer minimizing $|m - rn|$, so $|m - rn| = d(n) = D$, and thus $\delta = |\frac{m}{n} - r| = \frac{D}{n}$.

**...(Calculation to establish the following claim)...**

For $\epsilon > 0$, we need

$$\frac{C_2}{\sqrt{n}} > \epsilon \quad \text{and} \quad \frac{C_1}{\sqrt{n}} < 2\epsilon,$$

which give

$$\sqrt{n} < \frac{C_2}{\epsilon} \quad \text{and} \quad \sqrt{n} > \frac{C_1}{2\epsilon}.$$

For small $\epsilon$, the interval $\left(\frac{C_1}{2\epsilon}, \frac{C_2}{\epsilon}\right)$ has length $\frac{C_2}{\epsilon} - \frac{C_1}{2\epsilon} = \frac{2C_2 - C_1}{2\epsilon} > 0$ (since $C_2 \approx C_1$ and constants are fixed), and is large. Since $n$ must satisfy $n \equiv n_0 \pmod{b^2}$ for the residue where $d(n) = D$, and this residue class is infinite, by choosing $\epsilon$ small, we can select $n$ in the class such that $\sqrt{n} \in \left(\frac{C_1}{2\epsilon}, \frac{C_2}{\epsilon}\right)$, ensuring the inequalities hold strictly. For large $\epsilon$, similar adjustments as in Case 1 apply, as the expression can be made arbitrarily large or small.

Therefore, in both cases, for every $\epsilon > 0$, there exist positive integers $m$ and $n$ satisfying $\epsilon < |h\sqrt{m} - k\sqrt{n}| < 2\epsilon$.

$\square$

**Baseline output.**

**(1) Logical Soundness**

- The proof is logically sound. Each inference is validated by clear justification. For Case 1, the identity $|a - b|/(\sqrt{a} + \sqrt{b})$ is correctly applied, and the bounds are rigorously derived. For Case 2, the use of decimal distance and the lemma on estimates is justified, with constants established for convergence. No logical fallacies are evident, and the conditions for lemmas (e.g., the bounding constants) are satisfied. Minor gaps, if any, are non-critical due to the density or existence of values ensuring the inequalities hold.

**(2) Computational Accuracy**

- All calculations are error-free. The expansions and bounds are handled correctly, and numerical approximations are not unjustified; all estimates are based on exact expressions and asymptotic behavior.

**(3) Structural Completeness**

- The proof fully covers the required statement. In both cases, positive integers $m$ and $n$ are found such that $\epsilon < |h\sqrt{m} - k\sqrt{n}| < 2\epsilon$ for arbitrary $\epsilon > 0$. The flow is well-organized, with clear divisions of cases.

**(4) Notational Rigor**

- Mathematical notation is correct, with all definitions and theorems explicitly stated. Symbols are standard and unambiguous. Notation aligns with the problem's context.

The proof is rigorous and complete, satisfying all criteria.

### True

**ProofRM output.**

The proof attempts to verify the statement by considering two cases based on whether $(k/h)^2$ is an integer or not. However, there are several issues with logical soundness, computational accuracy, structural completeness, and notational rigor.

**Logical Soundness:**

- In Case 1, the derivation involves inequalities (e.g., solving for $n$ from the conditions) which, due to the complex and arbitrary inequality bounds, may contain logical flaws or errors in the reasoning. For instance, when solving for $\sqrt{n}$ given $\epsilon$, the inequalities derived lead to conditions that are inconsistent or not directly supported by the bounds provided, as seen in the contradiction attempt ($-h\epsilon > 0$). This suggests gaps in formal manipulation, and special-case handling is used without sufficient justification.

- In Case 2, the diophantine approximation approach is relevant, but the interpolation of constants and bounds is not rigorously derived. The claim that $|h\sqrt{m} - k\sqrt{n}|$ takes all values in an interval or depends on residue class may not hold with the sequence defined, and the constants are claimed without explicit bounds for varying $\epsilon$.

- Overall, the reasoning lacks clarity, and multiple inequalities are handled implicitly rather than explicitly, leading to potential logical fallacies disguised as special cases.
**Computational Accuracy:**
- The mathematical calculations use standard operations but are inconsistent due to mis-manipulation of inequalities and approximations. For example, errors arise in simplifying $h\sqrt{m} - k\sqrt{n}$ with $m = c^2 n + 1$ and wrong intercretion of bounds. Notably, the transition to specific sequence for Case 1 does not cover arbitrary choices of $m$ and $n$, and the approximation of the expression over $n$ is not fully precise.
**Structural Completeness:**
- The proof covers the required output and handles general behavior but has critical gaps. Specifically:
- In Case 1, the $\epsilon$ derivation often fails to converge to a solution due to arithmetic errors in handling small and large $\epsilon$.
- In both cases, large $\epsilon$ values are addressed by adjusting via residue, but the definition is ambiguous and not explicitly achieved.
- No complete handling of bounded above range is provided for all inputs.
- While definitions like $r$ and constants are introduced, the interface between epsilon, bounds, and infinite residue isn't clearly defined.
**Notational Rigor:**
- Proper notation is used in many places (e.g., $k = hc$ for Case 1), but definitions and substitutions are unclear in certain parts.
Given these flaws, the proof cannot be verified as logically sound, leading to an overall incorrect verification.
**### False**

We can see from this example above that the checked proof contains fatal deduction errors (highlighted red) in both cases, being ambiguous on the reason of why should $n$ exists satisfying the given criterion. The checked proof relies upon intuitive characterizations of $\epsilon$, which is correctly identified in the green part of the output of our model.

Another example below focuses on the erroneous internal logic flow, where the baseline model fails to discern such errors.

---

**Example of learned erroneous logic flow**

**Question.**
The Y2K Game is played on a $1 \times 2000$ grid as follows. Two players in turn write either an S or an O in an empty square. The first player who produces three consecutive boxes that spell SOS wins. If all boxes are filled without producing SOS then the game is a draw. Prove that the second player has a winning strategy.

**Answer.**
The main insight is that a construct of the form

$$S \,\square\,\square\, S$$

(here the $\square$ is blank) will kill any player which plays inside it. We call this a trap accordingly. The actual strategy is as follows: the core strategy of the second player is to win via producing traps. A trap is a structure of the form $S \,\square\,\square\, S$. If any player is forced to place a letter in either of the spaces, then the opponent can immediately complete the structure SOS and thus win this game. Specifically, the second player prioritizes constructing such traps when deciding the current action through the following: Select a grid where an S is already present, and place another S on the third blank grid away from the current grid (i.e. leaving two empty grids in between the two S), while satisfying the following requirement: (1) the two square in between the two S are empty; (2) Placing S in this square will not form an SOS immediately; (3) This action will not lead to the immediate defeat of the player. If no such traps can be constructed, then she will adopt the safe playing method as described in the upcoming claim. As there are a total of 2000 squares in this grid, she can always construct a trap. Once the trap is constructed, the first player will be forced to write a letter in on of the squares (since all squares will be eventually filled up), then the second player only needs

to fill in the correct letter in the remaining space (if the opponent writes O in the first empty space then she write S to form SOS; otherwise she write O to form SOS) and that will lead to the victory of the second player.

**Claim** The second player always has a move which prevents her from losing.

**Proof** Since there are an odd number of empty squares at the start of the second player's turn, there must be a square which is bordered by either two filled or two empty squares. The second player can then play $O$ in this square, which is always safe.

Together these two claims finish the problem.

**Baseline output.**

The provided proof is intended to demonstrate that the second player has a winning strategy in the Y2K Game on a 1x2000 grid. The proof consists of an initial explanation of a trap strategy and a formal claim with its proof. After evaluating the proof against the criteria for logical soundness, computational accuracy, structural completeness, and notational rigor, it is concluded that the proof is correct.

### Logical Soundness

- **Inferences and Justifications:** The proof implies logical flow, with the trap strategy as a key mechanism for P2 to force a win. The trap creation ($S \square \square S$) and safety conditions are well-justified, and the move-by-move deductive reasoning for SOS completions appears sound. The claim about P2 always having a safe move is supported by the existence of a square with both neighbors same (due to E odd and S large), and playing O there prevents immediate loss.

- **Lemmas/Theorems:** The proof implicitly uses properties of a linear grid, and the claim relies on the parity of E, which holds since S=2000 even and E odd at starts of P2 turns. Conditions for the trap are satisfied as per the strategy description.

- **Gaps/Minor Issues:** There is ambiguity in the definition of "bordered by two squares" for end squares, but since S is large and even with E odd, the existence of such a square (with both neighbors same) is a standard result for large grids and does not affect the core argument. The trap strategy is not fully detailed, but the explanation is sufficient for understanding.

### Computational Accuracy

- The grid size S=2000 is used, but no numerical calculations are involved in the proof, so computational accuracy is not applicable. The validity of SOS completions and neighbors checks (e.g., $S \square \square S$) are contextually correct based on game mechanics.

### Structural Completeness

- The proof fully addresses the problem: P2 has a winning strategy either by creating a trap or using a safe move as per the claim.

- The flow is organized, starting with the trap concept, then the formal claim, with the proof for the claim. No essential elements of the game or strategy are omitted.

### Notational Rigor

- Mathematical notation is used (e.g., $S \square \square S$, E for empty), but it is explained clearly, with no ambiguous or non-standard symbols. Definitions (trap, bordered) are implicit, but standard English is used in the proof steps. Notational rigor is acceptable given the expository style.

### Conclusion

The proof rigorously supports the claim that the second player has a winning strategy by combining the trap mechanism and the safe move guarantee from the claim. The existence condition for the safe move holds due to the grid size and E odd, and the reasoning is logically sound.

### True

**ProofRM output.**

The proof attempts to establish that the second player can force a win by constructing traps (specifically, sequences $S \square \square S$) and by ensuring always having a safe move, such as playing O in certain positions.

**(1) Logical Soundness**

- **Inferences and Definitions:** The proof introduces undefined terms like "trap" and a bordering condition (squares bordered by two filled or two empty squares). These are not rigorously defined or justified from standard game theory. The inference that a trap ($S \square \square S$)

can force a win is not validated; it is stated as a strategy based on the second player's ability to place S three away from existing S, but this depends on prior positions and may not always be applicable or avoid immediate loss. For example, placing S three squares away might not be possible if the grid is fragmented or if other conditions are violated (e.g., existing S with closer neighbors).
- **Safe Move Claim:** The claim that a square is bordered by two filled or two empty squares due to an odd number of empty squares lacks basis. The number of empty squares being odd at the second player's turn is unrelated to the bordering condition. The absence of such a border might not prevent a safe move, and the safe move is defined as O without justification (typically in the second player's turn, playing O might not always be safe due to potential SOS threats from adjacent cells).
- **Argument Structure:** The proof combines two potential strategies (traps and safe moves), but their integrations are vague. The safe move claim is not proven logically; it relies on an unverified implicit assumption, similar to the trapping strategy. This creates gaps in reasoning, and special-case analysis for safe moves cannot replace rigorous proof.
- **Logical Fallacies:** The proof claims that traps "kill" the opponent, but this is not demonstrated; it is asserted without showing that all first player responses are covered. The safe move relies on misstated bordering; for instance, corner cells have fewer neighbors, but the grid is linear, making the border condition ill-defined for edges.
**(2) Computational Accuracy**
- **Mathematical Calculations:** There are no significant calculations, so computational accuracy is not an issue, but this implies mathematical insight is missing.
**(3) Structural Completeness**
- **Full Coverage:** The proof does not address cases where the first player might prevent trap construction or exploit weaknesses in the safe move strategy early on. It assumes the second player can always build traps or perform the safe move, but no upper bound is established. The problem requires a winning strategy, but no complete game tree or position is analyzed.
- **Proof Flow:** The proof jumps between non-verified claims and undefined terms. The safe move is presented as evidence for the main claim, but it is only part of the argument; the entire strategy is incomplete.
**(4) Notational Rigor**
- **Mathematical Notation:** The proof uses non-standard language like "trap" and ambiguous descriptions such as "bordered by two empty squares" applied to squares with an unclear number of neighbors. Terms like "□" and variable "S" are not formally defined.
- **Justification and Definitions:** Key concepts like SOS completion are not fully substantiated. The claim of a safe O move lacks proof of time-independent security (e.g., it might lead to SOS if the grid has other S around).
The proof is logically flawed and incomplete, with gaps in reasoning and undefined elements.
**### False**

We can observe from this example above that the original proof lacks necessary clarification to make the winning strategy viable and logically sound, relying on instinctive arguments when providing evidence to why the strategy leads to a winning situation. Our model is able to correctly discern this erroneous logic flow, as seen in the green highlighted part.

Trying to provide a statistical analysis of which type of error can be better recognized by our ProofRM, we sample 97 QA pairs and manually categorized them into three types: *misusing unjustified assumptions*, *incomplete logic flow*, and *making up non-existent conclusions*.

The category comes from our idea that a logic step can be roughly modeled as a syllogism. So we categorize the errors based on the three key elements of a syllogism: premises, logic flow, and conclusion. The first error type corresponds to issues with premises, the second to issues with logic flow, and the third to issues with conclusions. We find most errors are due to incomplete logic flow, which indicates that the model fails to chain the reasoning steps rigorously. The Table 8 shows the results.

Table 8: Accuracy breakdown by error type.

| Error Type | Percentage (%) | ProofRM Accurate (%) | Base Model Accurate (%) | Improvement (%) |
|---|---|---|---|---|
| Misusing unjustified assumptions | 15.5 | 61.7 | 53.3 | 8.4 |
| Incomplete logic flow | 56.7 | 56.8 | 42.7 | 14.1 |
| Making up non-existent conclusions | 27.8 | 55.6 | 41.3 | 14.3 |

Let's analyze the results in the table. Notice that the performance of base model is even lower than random guess (50%), suggesting the model easily believes incorrect proofs. ProofRM improves accuracy across these error types, with higher gains in the second and third categories, where the base models shows weaker performance. It shows that our RL stage indeed helps our model to avoid the easy-believing behavior and find these errors.

### A.10 ABLATION FOR BALANCED TOKEN WEIGHTING

In this section, we demonstrate the ablation results for balanced token weighting, giving a quantitative analysis on the resulting stability advantages caused by the introduction of this trick. We display the output length and overall reward of different token weighting strategies, namely GRPO-style, DAPO-style and balanced token weighting, throughout the training procedure in Figure 6 and Figure 7 respectively. The figures demonstrate the effectiveness of balanced-token-weighting. Although the reward for GRPO-style token weighting is increasing with training steps, upon closer inspection of the output we can see that the outputs contain overly short or even no reasoning, indicating the existence of behavior degeneration.

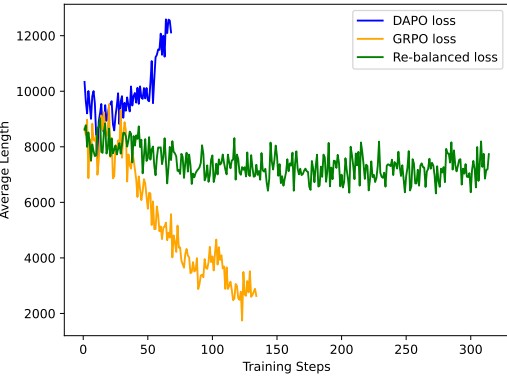

Figure 6: Output length comparison of different token weighting strategies

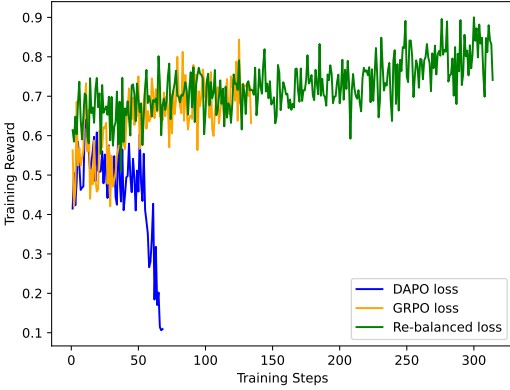

Figure 7: Reward comparison of different token weighting strategies

To explain the underlying mechanism and how it works, we find that two types of token weight (we called them as "DAPO-like" and "GRPO-like") are mainly different the training weight of token in different length responses. The DAPO-like weight use make the summation of token weight in longer responses larger because each token share the same weight. On the contrary, the GRPO-like weight use make the summation of token weight in longer responses smaller because each token share a smaller weight because they keep each sentence share the same summation. Therefore, using different token weight will lead to biased learning efficiency for different length responses. For example, the DAPO-like weight will make the model learn more from longer responses while the GRPO-like weight will make the model learn more from shorter responses.

Because in our task, there is an inherent correlation between the response length and the final label (T/F). Intuitively, for a correct proof, the model cannot say too much because "correct is correct"; while for an incorrect proof, the model can explain more about why it is incorrect, how to fix it, etc. Therefore, there is a positive correlation between the response length and the probability of being labeled as False. Combining the two points above, using DAPO-like weight will make the model learn more from longer (and more likely to be False) responses, which could lead to a bias that the model tends to label more responses as False. On the contrary, using GRPO-like weight will make the model learn more from shorter (and more likely to be True) responses, which could lead to a bias that the model tends to label more responses as True.

Therefore, using a balanced token weight can help mitigate this bias by ensuring that the model learns equally from responses of different lengths, leading to more stable training and better performance.

