# OpenReview forum: "ProofRM: A Scalable Pipeline to Train a Generalized Math Proof Reward Model"
_ICLR.cc/2026/Conference — Submitted to ICLR 2026_

### Official Review · Reviewer_BDhi · 2025-10-31

**Soundness:** 2
**Presentation:** 2
**Contribution:** 3
**Rating:** 4
**Confidence:** 3

**Summary:**

This paper addresses the challenge of verifying mathematical proofs in reinforcement learning settings. The authors argue that current RLVR (Reinforcement Learning with Verifiable Rewards) methods, which work well for problems with easily verifiable answers, struggle with proof-based problems where verification asymmetry is weak. They propose a data collection pipeline that generates diverse Question-Proof-Check (QPC) triplets through multi-dimensional LLM-aided generation, followed by LLM labeling with hierarchical human validation. The collected data is used to train a generative reward model with stabilization techniques including "LLM-as-RM-for-RM" and balanced token weighting. Experiments show the model achieves comparable performance to strong baselines on proof verification tasks.

**Strengths:**

1. **Well-motivated problem**: The paper clearly articulates the limitation of current RLVR approaches for proof-based mathematical problems, providing concrete evidence that verification asymmetry breaks down for complex proofs.

2. **Comprehensive data collection effort**: The multi-dimensional diversity approach (question sources, generation methods, LLM diversity) is sensible and generates a substantial dataset. The composition-level human validation strategy offers a practical trade-off between scalability and quality.

3. **Practical engineering contributions**: The LLM-as-RM-for-RM mechanism and balanced token weighting are pragmatic solutions to real training instabilities. The stable training over 312 steps with 18k samples demonstrates the effectiveness of these techniques.

**Weaknesses:**

1. **Limited novelty**
   The work mainly integrates existing components — multi-LLM data generation, consistency filtering, RLVR-based reward modeling — into a single pipeline.
   Conceptually it remains an **engineering consolidation** rather than a new methodological advance; the two stability heuristics are heuristic extensions without theoretical grounding or extensive empirical validation.

2. **Insufficient ablations and analysis**

   * No ablation for different data diversity dimensions, RM-for-RM or balanced token weighting.
   * No comparison between **base vs. trained ProofRM** on test-time scaling datasets.

3. **Weak test-time scaling / relative ranking ability**
   Although ProofRM performs well on single-sample accuracy, its *best-of-k* curve in Fig. 4 quickly flattens and stays below GPT-5-mini and Gemini.
   This indicates **poor calibration and limited ability to rank multiple diverse proofs**, suggesting that the model learns binary classification but not fine-grained relative judgment.
   The small and filtered evaluation set (18 problems, 48 proofs) further limits statistical reliability.

**Questions:**

While ProofRM shows strong single-sample accuracy in §4.1, its performance scales poorly in §4.2 (best-of-k).
Could the authors provide a deeper analysis of why good point-wise classification does not translate into effective relative ranking across multiple candidates?
Understanding whether this gap arises from binary supervision, calibration issues, or biases introduced by the stabilization methods would greatly clarify the behavior of the reward model.

---

> ### Author Response · Authors · 2025-12-03
> **Reply 1/n: novelty**
>
> 1. **W1**, **Novelty**:
>     - Our paper is the first paper to provide a training set to train a proof verification model, which has long been a missing piece in the LLM-for-math literature. Our data generation pipeline is novel and one of our main contribution, which is the key to make it possible to automatically data generation with less human check. Compared to some related work listed in our paper like OPC and even very recent ProofBench, the authors only create a benchmark and they highly dependent on human efforts, which limits their scalability to extend the benchmark to a training dataset.
>     - Our combination-level human check is based on our multi-dimension diversity design. If we do not have these combinations, the human check must be accept or reject all samples. So our method is a novel and clever trade-off between human effort and data quality. This means our pipeline is a highly integrated, cohesive system.
>    - Our techniques are based on our detailed analysis of training dynamics and provide some valuable insights for future research. For example, our LLM-as-RM-of-RM is based on the significant weakening of the correlation between process quality and the correctness of the final result in the binary classification case. Our balanced token weight is also based on our insights about the length bias related to the task nature. We have also added some theoretically analysis (see Point 4 for Reviewer `S5u1` and Point 8 for Reviewer `WK1o`). We have included them into the Appendix A.8 and A.10 in the revision. We are also happy to further extend some of our empirical techniques to more theoretical analysis in the future.
>     - Finally, we want to emphasize that our main contribution is a practical recipe. Engineering consolidation work is also an important contribution for the community, especially in the era of LLMs where the importance of data and scalability has been more and more showed.

---

> ### Author Response · Authors · 2025-12-03
> **Reply 2/n: More ablation and comparison**
>
> 2. **W2**, **Ablation and comparison**:
>    - Thank you for your suggestion. We have added a leave-one-out ablation of the diversity dimensions. See point 3 in response to `WK1o` for ablation about diversity dimensions. As for LLM as RM for RM and balanced token weighting, we newly provided details and analysis on it. See Appendix A.10 for more detailed ablation analysis about their effectiveness.
>    - We have added some new benchmarks (like OPC, ProofBench) and comparison between base vs trained ProofRM. See point 1 of reviewer `S5u1`. Our ProofRM also shows improvement on these benchmarks without specially training on their style.
>    -  We have added the baseline in the test-time scaling experiments, showing that better test-time scaling performance of ProofRM.
>
> 3. **W3, Q1**, **Test-time scaling performance**:
>     - First, sorry for our misleading description. Our initial best-of-k results are based on 18 problems and 48 proofs for **each problem**. So the total QA pairs are 864 instead of 48.
>     - Then, we have **updated this experiments**. Our new results show better performance of ProofRM than the baseline and some closed-source LLMs, suggesting good test-time scaling ability. Specifically,
>         - We find that the current not good performance of our best-of-k curve does not mean real worse ability but some minor bug in our  test set. We find some of test samples in our best-of-k set have really poor performance like very short proof without meaningful proof process. Our other training set and test set do not include these *'trivial negative proofs'*, because they are too naive. These samples are only included in the best-of-k test set, which is mainly imported by the Qwen, which is significantly weaker than others. We use the model to provide a large range of proof diversity to test the *"ranking ability"*. However, we believe these too short or even almost "*empty*" samples are somehow too naive to be used as a testbed.
>         - Our model has not been trained on such proofs. We find that on these problems, our model refuse to provide a True or False answer but complain the input is incomplete, affecting the performance of our model.
>         - To provide a more accurate estimation of the test-time scaling ability, we refine our test set to remove those samples of too low performance, and also debug for our answer matching to correctly solve these cases. Our new experiments show that our model show good test-time scaling performance.
>        - To further solve the bias, we also added some *"odd"* proofs like empty or very short proofs or proofs with nonsense content in the training data to help our model to better handle these samples. With these data samples, our new 8B model, 14B model and 32B models can correctly solve these samples.

---

### Official Review · Reviewer_S5u1 · 2025-11-01

**Soundness:** 2
**Presentation:** 2
**Contribution:** 2
**Rating:** 2
**Confidence:** 3

**Summary:**

This paper presents ProofRM, a reward model designed to verify the correctness of mathematical proofs expressed in natural language. The authors propose a three-stage pipeline: (1) automated data collection using LLMs with multi-dimensional variation (problem sources, proof generation methods, and generating LLMs), (2) LLM-aided labeling with composition-level human check, and (3) RLVR training using a generative reward model "LLM-as-an-RM-for-RM". The trained ProofRM achieves high accuracy on diverse test problems and demonstrates effective test-time scaling capabilities with comparable performance on best-of-k metrics.

**Strengths:**

The paper identifies an important and timely problem: the asymmetry of verification for mathematical proofs. The authors convincingly argue that while answer-checking has enabled breakthroughs in mathematical reasoning (DeepSeek-R1, OpenAI o1), proof verification remains a bottleneck for Olympiad-level and research-level mathematics. A comprehensive and scalable data collection pipeline is a major strength. The multi-dimensional diversity strategy including question source diversity, proof generation method diversity and LLM diversity seem sound. The composition-level human validation is quite elegant, requiring only ~100 hours of human effort versus >1,200 hours for exhaustive annotation. The paper also introduces “LLM-as-an-RM-for-RM”: an auxiliary LLM detects surface-level fluency issues (repetition, context-irrelevance, hasty conclusions) to prevent positive rewards for accidentally correct but flawed reasoning.

**Weaknesses:**

__1) Evaluation Limitations__

__a) Benchmark construction concerns__

The experimental section includes two types of problems: generalized check accuracy and test-time scaling.
In the first setup, the "Mix-up" test set comes from the hierarchical sampling used during data construction. This creates a substantial bias as the baselines considered are not familiar with the dataset. While the paper states these are "different questions," they come from the same source distributions and LLM generations as training data. The second set (“student-sourced problems”) is not discussed at all, and none examples are presented. These two observations about test data sources raise concerns about the fairness of the entire evaluation in this subsection.

The second setup considers test-time scaling evaluation on 18 problems. This is an extremely small set as the total number of the generated proofs is 48. The confidence interval for such an evaluation is likely wide. The paper should at least report statistical significance testing and, potentially, increase the number of problems considered.

__b) Limited comparison to existing approaches__

The paper provides no baseline against existing proof RMs. The authors mention "Open Proof Corpus" with ~5k annotated samples but provide no head-to-head comparison.

The paper briefly mentions MathConstruct, IneqMath, and DeepTheorem but dismisses them based on a single author's informal experiment (Appendix A.3). This analysis lacks rigor: success rates of 84%, 70%, and 40% by a non-expert within minutes suggest these datasets may not be "fundamentally altered" but rather require targeted defense mechanisms.

__c) No computational overhead analysis__

The paper does not report the inference cost. Generating reasoning traces for each proof during training/evaluation incurs substantial computational overhead. How does ProofRM's throughput compare to ORM/PRM baselines? For test-time scaling with k=50, does the 8× repeated generation (64 total generations per proof) remain practical? The absence of comparison to discriminative RMs, especially time-comparison, does not benefit the work.

__2) "LLM-as-an-RM-for-RM" Approach Justification__

While the "LLM-as-an-RM-for-RM" approach addresses real problems, its design feels reactive rather than principled. Why should surface-level fluency checks prevent collapse, and could this inadvertently penalize valid but unconventional reasoning styles? The authors provide post-hoc manual inspection (50 samples), however, some statistically significant validation is required. Additionally, Figure 3 shows reward increasing for 18k samples, the paper acknowledges "model collapse still occasionally occurs". However, the paper provides minimal qualitative analysis of failure modes. The discussion in Appendix A.6 provides negative examples but no systematic characterization or ablation study.

__3) Overstated Scalability Claims__

The paper claims the approach is "scalable" in multiple places, but there are major bottlenecks in the pipeline. First, human validation is still required. The hierarchical sampling saves effort but doesn't eliminate human involvement. For new problem domains, the 100-hour investment must be repeated. This doesn't scale to diverse mathematical fields (e.g., topology or differential geometry). Second, the pipeline relies on the LLM-as-a-judge approach in multiple places (proof generation, labeling). This limits the approach to the tasks and domains that are LLM-friendly. The approach cannot handle diagrams, symbolic computation, or formal language.

__4) Missing Related Work__

The paper does not discuss the outcome reward models vs. process reward models in the context of proof verification. PRMs provide step-level supervision—could this approach be combined with ProofRM?​​Limited engagement with test-time compute scaling literature beyond best-of-N. Techniques like beam search with PRMs or lookahead search could enhance ProofRM's utility.

**Questions:**

1) Can you provide a direct comparison to OPC and other proof verification baselines on a standardized (preferably different than the one presented in the paper) test set?
2) Have you experimented with discriminative (scalar) reward models? What is the accuracy-efficiency (especially time efficiency) trade-off compared to the generative approach?
3) How does the optimal token weight η vary across different model sizes (1.5B, 3B, 8B)? Is 0.6 universally optimal?
4) Could the pipeline be adapted to formal language (Lean/Isabelle) by replacing LLM proof generation with auto-formalization? What would be the cost-benefit trade-off?
5) How does ProofRM performance scale with training data size? Is there a clear saturation point, or do you expect continued improvement with 100k+ samples?

---

> ### Author Response · Authors · 2025-12-03
> **Reply 1/n: Evaluation Limitations**
>
> 1. **W1.1**, **Benchmark diversity**:
>    - Thank you for your suggestions, to further reduce the interference of biases in our test set, we have added more benchmarks like ProofBench and OPC to our test set and also added them in the revised draft. The details of our newly enlarged test set is already updated in Appendix A.6. A detailed analysis paired with examples about the student style proofs is already provided in our first draft in Appendix A.4.2.
>     | **Model** | **OPC** | **ProofBench** |
>     |---------|-------|--------------|
>     | Base-8B | 70.8 | 59.3 |
>     | ProofRM-8B | 75.7 | 67.5 |
>     | Base-14B | 69.4 | 59.8 |
>     | ProofRM-14B | 77.4| 68.5|
>     | Base-32B | 70.2 | 56.9 |
>     | ProofRM-32B | 81.3 | 69.4 |
>
>         These experiments show our model can generalize to other benchmarks. Notice that we find our base models (Qwen-3) does not show that better accuracy with larger size. We believe that it is because the task is significantly hard for the base models. But our ProofRM can indeed achieve better performance.
>    - The mix-up partition of the test set is mainly to demonstrate the in-domain performance gain, supporting our gain from the stable RL.
>    - We are sorry for our misleading statement (*48 proof test set*). The test-time scaling dataset is on 18 problems and 48 proof for **each**, so the amount of total QA pairs is 864. For more details, see point 1.2 for reviewer `WK1o` and point 3 for reviewer BDhi.
>
> 2. **W1.2**, **Q1**, **Comparison** to existing approaches:
>    - We appreciate your opinion, and we have now provided the comparison to the Open Proof Corpus in the revised draft.Here, our model shows comparable performance (81.3% v.s. 83%) where they use an in-domain RM but we do not focus on their distribution.
>    - We would still like to point out that the Open Proof Corpus is proposed within 3 months of our submission, and should thus be considered as a concurrent work.
>    - Additionally, we admit that OPC is a good work where the authors have done a lot of human effect to annotate the data, but consequentially these data are **limited** and **hard to further scale up**, which is the main contribution of our work. Compared to OPC, our work 1) focus on provide a more diverse and larger-scale dataset, where we finally get 33k+ QA pairs and can be further scaled up by introduce new LLMs or datasets. 2) In our pipeline, we try to get a clever trade-off between less human effort (so that can be scale up) and high-quality annotation (so that the data is useful).
>    - For methods like MathConstruct, IneqMath and DeepTheorem, we admit that some claim could be not rigorous enough. Thank you for your reminder. We have revised the related description. Our experiments in Table 2 are mainly to explain our motivation that existing methods trying to create verified results for proof could be vulnerable and deviate from our goal of ensuring the correctness of complete logic flow of proofs. We are also expecting some further improvements in this direction that can improve the defense ability.

---

> ### Author Response · Authors · 2025-12-03
> **Reply 2/n: Inference cost**
>
> 3. **W1.3**, **Q2**, Inference cost
>    - **Report of inference time**: In our experiments, we use a 8 GPU node to inference with vLLM. Our ProofRM-8B check 283 QAs with 4 times check (4*283=1,132) in one hour, which is acceptable for this level math problems (Olympiad-level). Notice that considering the complexity of these problems, using a reasoning model as RM is the common (and even default) method. See the report of the [Gemini-IMO](https://arxiv.org/abs/2507.15855) and [DeepSeek-Math-v2](https://arxiv.org/html/2511.22570v1).
>     - **Accuracy efficiency** trade-off: Our choice of **generative RM** is based on the fact that the GRM has better generalized accuracy, which is one of the most focused thing in our paper. To support it, we use the same training set to train a model only output the T/F, without any reasoning and error report, called *"Scalar ProofRM"*. The inference speed is about 40% of the generative RM, while we find the prediction of the scalar ProofRM is unreliable. We find that it is highly prone to reward hacking during training: the policy can exploit shallow, goal-irrelevant cues (e.g., *proof length*, *linguistic style*, or *templated phrasing*) to obtain high reward, rather than improving the intended capability. Consequently, though it works for in-domain setting, it generalizes poorly under OOD settings (see Point 5 for reviewer `WK1o`), raising concerns about its reliability as a supervision signal.
>         | Model | In-domain Accuracy | OOD Accuracy |
>         | --------| ----------------------- | ----------------- |
>         | Base Model | 71.60 | 62.3 |
>         | Generative RM | 76.81 | 64.7 |
>         | Scalar RM | 83.12 | 47.3 |
>     - **Trustful output**: Moreover, a scalar score provides little interpretability or actionable evidence for auditing which further undermines trust and makes “checking the RM” substantially harder. In contrast, a generative RM can provide human-readable criteria or structured critiques, enabling more effective verification and debugging; therefore, we consider it a more reliable option in practice.
>     - **Test-time scaling**: A similar and important topic is **test-time scaling**, where how can a model improve its performance with more inference time is also an important metric. This is what our best-of-k experiments show. We find a scalar RM cannot benefit from more reasoning attempt and shows a flatter curve, restricting its usage for harder problems.
>     - **Summary**:  As a conclusion, if a user only focuses on train an RM which is only used in in-domain setting, we believe it is acceptable for a scalar RM but our data pipeline and dataset are also useful. If the user also focuses on some generalized ability or interpretation, the generative RM is a better choice.
>     - **Repeated generation is practical** because 1) the repeated generation is highly parallelizable 2) it is commonly used in current math system. In our experiments, generating 48 times generation only cost about 1 hours for 8 GPUs. For some more modern systems, they will use significantly larger repeating times like 128 or 512 to further improve the performance. Also see the report of the [Gemini-IMO](https://arxiv.org/abs/2507.15855) and [DeepSeek-Math-v2](https://arxiv.org/html/2511.22570v1).

---

> ### Author Response · Authors · 2025-12-03
> **Reply 3/n: "LLM-as-an-RM-for-RM" Approach Justification**
>
> 4. **W2**, Justification of **"LLM-as-RM-of-RM"**:
>    - The collapse we observed in our training is usually due to some incorrectly encourage introduced by these ''surface-level'' fault. We observe that in some steps, some checking generation will include some repeated phrases or words (like ''so so so so'') but finally leads to a correct T/F judgment. If our RL process incorrectly encourage these samples, after several steps, the model may learn to generate repeated words more and more frequently, which finally lead to the collapse. So it is important to find these unexpected behaviors as early as possible and avoid encouraging them. In our training stage, we find at some steps, the odd generation occurred, and after 5-10 steps the entropy will significantly explode and the reward will drops to almost 0.
>    - Our LLM-as-a-judge-for-RM focuses on some basic language problems. Our target is to penalty some sequence like repeated "so so so so" or odd Chinese characters or Arabic characters in English environments. Even for some unconventional reasoning steps, the generation should be also the fluent language sequence, so it will not be penalized incorrectly. The high consistent rate displayed by post-hoc human checks(> 96%) also strengthens our confidence that the model's reasoning ability is not compromised.
>    - From a statistical perspective, deriving from the 96% TN and TP rates of post-hoc human checks, simple calculation shows that this approach yields an F1 score of 0.96, indicating a high level of reliability in distinguishing correct from incorrect proofs. This statistical evidence supports the effectiveness of using LLM-as-RM-of-RM in our training process.
>    - We sincerely appreciate your suggestion about analysis of failure modes, thus we will here continue our discussion already present in Appendix A.6 (now Appendix A.8 in the revised draft). It is worth noting that the surface-level fault that we minimize with the deployment of LLM-as-RM-of-RM is a characteristic of collapse, rather than the cause of collapse itself. The most direct indicator, or systematic characterization, is the sudden drop in accuracy to **nearly zero** and the concurrent variance in output length. So this RL must be stopped and restart from its recent checkpoint. So it cannot be analyzed in a statistical style. And as is discussed in Appendix A.8, we have observed that surface-level faults often precede these more direct indicators of collapse. Therefore, by addressing surface-level faults early, we can effectively prevent the onset of collapse in the training process, thus further enhancing the stability of our RL training.

---

> ### Author Response · Authors · 2025-12-03
> **Reply 4/n: Scalability**
>
> 5. **W3**, **Scalability**:
>    - As our best known, our data pipeline achieve the best scalability while ensuring the data performance and avoiding easily attack (like these methods in Table 2). Specifically, ...
>    - Human effort and performance is a fundamental trade-off and our contribution is to provide a method to get a better trade-off. Our method use a significantly less human effort (**only 1/12**) compared to one-by-one check compared to some other works like OPC. To collect 20k data, we only use 100 hours of human time, which just requires several annotator to finish in several days. The cost is acceptable for most research labs. So we believe it is scalable enough.
>    - We admit that our methods is mainly based on some basic LLM capabilities and could be not directly applicable to some domain. But first, natural language math proof is an important domain itself and we never claim our method can be applied to all domains. Then, for your mentioned examples like diagrams or symbolic reasoning, LLMs are also rapidly improving (for instance [Gemini-3-pro](https://blog.google/products/gemini/gemini-3) and [Deepseek-V3.2](https://huggingface.co/deepseek-ai/DeepSeek-V3.2/blob/main/assets/paper.pdf)) and a similar pipeline could be applicable in these domains in the near future.

---

> ### Author Response · Authors · 2025-12-03
> **Reply 5/n: Related works and Other questions**
>
> 6. **W4**, **Related works**:
>    - See point 2 in response to WK1o.
> 7. **Q3**, **Choice of** $\eta$:
>    - It is not universal but related to the models' prior behavior and dataset distribution. But as long as the training is stable, the final performance is not sensitive for the $\eta$. At the same time, an inappropriate $eta$ could lead to length collapse just after tens of RL steps so it is not hard to tune it in practice. In our experiments to scale up the model size, we adopted the constant $\eta=0.6$ for model size 8B, 14B and 32B, and all the training processes are stable.
> 8. **Q4**, **Formal language**:
>    - Formal language's most benefit is its verifiability and its drawback is significantly fewer training data and less readable format. We do not need an RM for formal language because the correctness can be directly verified. More discussion can be found in Point 4 for Reviewer `WK1o`.
> 9. **Q5**, **Scaling with training data**:
>     - As long as the training is stable, we find more training data and more RL steps will not harm the performance. There exists some saturation effect where after some point the performance gain become smaller. And the point will be later if the model size is larger. For our original 8B model, it roughly about 13k. In our newly added experiments about 14B and 32B models, the saturation point is further later which is 22k and 26k. (We acknowledge it is not a universal point but specific in our learning rate and batch size setting.) So we believe with larger models, further scale up of training data and RL steps could further improve the performance.

---

### Official Review · Reviewer_WK1o · 2025-11-05

**Soundness:** 2
**Presentation:** 3
**Contribution:** 2
**Rating:** 4
**Confidence:** 3

**Summary:**

This paper addresses the challenging problem of verifying mathematical proofs in reinforcement learning context for LLMs. The authors argue that although RLVR performs well for problems with easily verifiable answers, the proof-based problems are lacking this verification asymmetry. They propose a scalable pipeline for creating question-proof-check (QPC) triplets by multi-dimensional diversity expansion and hierarchical human review, then train a generative reward model (ProofRM) which achieves 76.8% accuracy on diverse proof evaluation tasks.

**Strengths:**

Originality: The paper identifies important gap in current RLVR approaches - that verification asymmetry breaks down for proof-based problems. The multi-dimensional diversity expansion approach (varying sources, generation methods, and different LLMs) combined with composition-level human checking is creative solution for scalable data collection.

Quality: Technical contributions are quite solid, including "LLM-as-RM-for-RM" mechanism for preventing training collapse and the balanced token weighting for stabilizing output length. The detailed analysis of negative sample types in Section A.4.4 gives valuable insights about common proof failure modes.

Clarity: The motivation is articulated well with concrete examples in Table 1 and 2. Pipeline is illustrated clearly in Figure 1, and implementation details are documented thoroughly in appendix.

Significance: This work addresses practical bottleneck in scaling of mathematical reasoning - the lacking of scalable proof verification. The reported 12-fold efficiency gain in annotation process and successful RL training over 300+ steps show practical value.

**Weaknesses:**

The experimental scale appears limited with training set of only 24k samples and test set containing just 18 problems for best-of-k evaluation, which are too small to make robust conclusions. The 8B parameter base model is also limiting the generalizability claims. There is insufficient comparison with baseline methods - no comparison with process reward models (PRMs) or MCTS-based approaches except brief mentions, and missing the ablations on individual contributions of diversity dimensions.

The evaluation has several gaps including no evaluation on formal theorem proving, research-level mathematics, or out-of-distribution proof styles. The "Student" test set is showing significant performance drop (47.18% vs 76.81% on Mix-up), which suggests limited robustness. Methodologically, the heavy reliance on LLM-generated labels even after filtering could introduce systematic biases, and the balanced token weight formula is lacking theoretical justification. Also composition-level checking maybe misses instance-level errors.

**Questions:**

How does the performance change with dataset size? Could you provide learning curves that show relationship between training data volume and accuracy? What is inter-annotator agreement for human checks and how many disagreements were occurred in the 5% sampled questions?

What patterns ProofRM learns for identifying incorrect proofs? Can you show attention visualizations or feature analysis about what model focuses on? You mention briefly MCTS-based PRMs - could you provide quantitative comparison or explain why your approach is fundamentally different or better?

What is computational overhead of LLM-as-RM-for-RM during training and how this affects the training efficiency? Have you tested ProofRM on formal proofs (like Lean/Isabelle) or research-level mathematics beyond the competition problems? Could you provide statistics about which error types (from Section A.4.4) ProofRM catches most effectively and least effectively?

---

> ### Author Response · Authors · 2025-12-02
> **Reply 1/n: further scale up**
>
> 1. **W1.1**, **Need further scale up**
>    - We agree that further scale up is important and we are now working on it. About your mentioned three aspects:
>       1) *"24k training set is limited"*
>          - Our data pipeline is design to generalizable and can be further scaled up with more **generating LLMs**, **data sources**and **prompt methods**. Indeed, we have enlarged our training set from **24k** samples to **33k** samples by adding some new prompt methods. At the same time, the most easy way to further scale up is to introduce more generating LLMs. The current number is enough for our experiments so we have not further introduced more LLMs considering the cost. But with more LLMs introduced, the data size can be further scaled up.
>          - 33k samples are already a large dataset for RL finetuning in one field (Olympiad-level math proof). Some related works like OPC [1] only have 5k samples and the RLVR-MATH from allenai have 7.5k samples.
>       2) *"test set containing just 18 problems"*
>          - We are so sorry for our misleading description (*"48 proofs"*). Our "test-time scaling" testset contain 18 problems but 48 proofs for **each problem**. So the total input pairs are 864, which is not a small **test set** compared to some related work [1-2]. As a comparison, the "best-of-n" test set in OPC contains 480 QA pairs and the generic test set contains 676 QA pairs. We have clarified this in the revised draft.
>        3) *"8B model is limited."*
>          - We newly RL finetune different model sizes including 14B and 32B models. Our RL data and recipe can consistently stably improve the performance with different model size. These different models all show significant improvement over their baselines. At the same time, the larger models show better improvement because of their stronger learning ability, which aligns with the *"scaling law of RL"* [3]. The results are shown below.
>
>             | Model Size | Base Model   | ProofRM   |  Improvement |
>             |------------|--------------|-----------|--------------|
>             | 8B         | 71.6%        | 76.8%     | 5.2%         |
>             | 14B        | 71.9%        | 79.0%     | 7.1%         |
>             | 32B        | 72.5%        | 82.4%     | 9.9%         |

---

> ### Author Response · Authors · 2025-12-02
> **Reply 2/n: More comparison**
>
> 2. **W1.2, Q2.2**, **Compare to more baselines**, like PRMs, MCTS-based approaches.
>    - The ORM is the first choice because of its stability and simplicity. Especially for math proof, the step value or the process reward is hard to define and could be very noisy. For example, in some problems, the value of a reasoning step cannot be determined until the final proof is finished. If a latter step is wrong, we cannot determine whether the current step is valuable or not.
>    - However, our pipeline can be further extended to PRM. In fact, in our data generation process, the LLM votes also contain step-level evaluation information, which can be further extracted to get PRM data.
>    - Similarly, the MCTS-based approach needs an answer reward to estimate the node value. This paper focus on construct the problem-proof-T/F dataset from the scratch and verify the effectiveness of our data pipeline. Our ORM is the first step before trying more complex methods like MCTS. It is a valuable future work to further explore more complex RM.
> 3. **W1.3**, **Ablation study**: *"needs ablation about each diversity dimension."*
>    - We added some ablation study about the diversity dimensions. Here, we operate **leave-one-out ablation** experiments, where we remove one diversity dimension each time and retrain the ProofRM with the remaining data. The results are shown below. Here all the models are the same of our main experiments except the training data. The results show that **each dimension contributes to the *diverse data quantity* and also *the final performance***. Specifically, the diversity introduced by the ***"prompt method"*** is the key to train a generalized model, where the model without this diversity shows worse accuracy than its base (69.9% v.s. 71.6%). Here, the base model has accuracy 71.6% and our full ProofRM has accuracy 76.8%.
>
>         | Removed Dimension        | ProofRM Accuracy | Data ratio retained | Performance Drop v.s. Full|
>         |--------------------------|------------------|---------------------|------------------|
>         | w/o Dataset Source       |     72.9%      |      43.97% |        3.9%      |
>         | w/o Extended LLM         | 74.1%              | 38.71%                 |       2.7%        |
>         | w/o Prompt Method        | 69.9%              | 32.20%                 |         6.9%       |
>
>       Specifically, removing the dataset source dimension, we only keep the data from *OlymBench*; removing the extended LLM dimension, we only keep the data generated by *deepseek-R1*; removing the prompt method dimension, we only keep the *rephrase* part.

---

> ### Author Response · Authors · 2025-12-03
> **Reply 3/n: more OOD evaluation**
>
> 4. **W2.1**, Evaluation on **formal theorem**
>     - Formal theorem proving is a essentially different direction from natural-language proof. The formal language proof has its own proof style, tools and the proof is often hard to understand by human. An LLM is trained to read natural language proofs is not directly applicable to formal theorem proving.
>     - Similarly, formal theorem proving model cannot directly solve natural language math problems or review natural language proofs. The translation must be done by some special models or human experts, while the LLM translation performance is still limited. See some related works like [4-6]
>     - As we discussed in Section 5.1, the formal language proof is an important direction but is still far from natural language proof in many aspects. The main weakness is the very limited training data, biased corpus domain (for example, the formal language is hard to represent Mathematical Analysis or Combinatorics) and the hard-to-used format. We acknowledge the importance of formal theorem proving but the natural language proof itself has been generally considered as a promising direction like [Gemini IMO solve](https://arxiv.org/abs/2507.15855) and very recently released [DeepSeek-Math-V2](https://github.com/deepseek-ai/DeepSeek-Math-V2/blob/main/DeepSeekMath_V2.pdf).
> 5. **W2.2**, Research-level mathematics and out-of-distribution proof styles:
>     - **Research-level mathematics**: we find that no test set of research-level math proof can be publicly available now and collecting such data require too much human effort. We cannot find qualified human experts to annotate a test set where some correct and incorrect proofs should be categorized. However, we are glad to provide some case study in Appendix.
>     - **Test on OOD proof styles**.
>       - Our models have tested on LLM-generated style, official solution style (both in Mix-up set) and student proof style (in Student set). Our student style test has shown that the performance on the OOD style, which is significantly different from our training data style. The results show that our model still shows improvement over its baseline (44.36% vs 47.18%). More explanation is found in point 6.
>       - We also construct some other OOD proof styles by LLM style-transfer technique. For example, we use an LLM to translate the original proof to different styles including:
>         - Math website (like Art of Problem Solving) with community solutions (conversational tone).
>         - Math textbooks. (More formal, many definition, lemma etc.)
>         - Math research articles. (assume strong background knowledge, minimal hand-holding)
>       Then we test our trained ProofRM on these different styles. The results are shown below.
>
>         | Proof Style               | Base Model Accuracy | ProofRM Accuracy | Improvement |
>         |---------------------------|---------------------|------------------|-------------|
>         | Math Website Style       | 56.7%                 | 58.2%              | 1.5%         |
>         | Textbook Style           | 65.2%                 | 68.5%              | 3.3%         |
>         | Research Article Style   | 65.0%                 | 67.5%              | 2.5%         |
>
>         The results show that our model can generalize to these OOD proof styles and consistently improve the performance over its base model.
> 6. **W2.3**, Performance on **student styles**
>     - The performance drops mainly because of the harder nature because less detailed steps and more gaps notice that minor gap is generally considered acceptable when coach scores student solutions but larger gaps are not allowed. Notice that the base model also shows a significant performance drop (71.36% vs 44.36%).
>     - The fair comparison should be the improvement over the baseline. The mixup improvement (71.6% -> 76.8%) and the student style improvement (44.36% -> 47.18%) are comparable. It shows that our model can generalize to these harder styles.

---

> ### Author Response · Authors · 2025-12-03
> **Reply 4/n: Theoretical explanation of re-balance token weight**
>
> 8.  **W2.5**, Theoretical explanation of why balance token weight is useful.
>    - Because algorithm modification is not the main contribution of this paper, we only provide a direct solution without theoretical analysis in the paper. We indeed have some preliminary "theoretical" analysis about why balanced token weight can help avoid length collapse and stabilize the training.
>    - We believe that the problem is base on some inherent correlation in the dataset.
>      - First, two types of token weight (we called them as "DAPO-like" and "GRPO-like") are mainly different the training weight of token in different length responses. The DAPO-like weight use $1/\sum |o_i|$ make the summation of token weight in longer responses larger because each token share the same weight. On the contrary, the GRPO-like weight use $1/|o_i|$ make the summation of token weight in longer responses smaller because each token share a smaller weight because they keep each sentence share the same summation.
>      - Therefore, using different token weight will lead to biased learning efficiency for different length responses. For example, the DAPO-like weight will make the model learn more from longer responses while the GRPO-like weight will make the model learn more from shorter responses.
>      - Because in our task, there is an inherent correlation between the response length and the final label (T/F). Intuitively, for a correct proof, the model cannot say too much because "correct is correct"; while for an incorrect proof, the model can explain more about why it is incorrect, how to fix it, etc. Therefore, there is a positive correlation between the response length and the probability of being labeled as False.
>      - Combining the two points above, using DAPO-like weight will make the model learn more from longer (and more likely to be False) responses, which could lead to a bias that the model tends to label more responses as False. On the contrary, using GRPO-like weight will make the model learn more from shorter (and more likely to be True) responses, which could lead to a bias that the model tends to label more responses as True.
>      - Therefore, using a balanced token weight can help mitigate this bias by ensuring that the model learns equally from responses of different lengths, leading to more stable training and better performance.
>      - **Summary**: Here, we do not claim that the re-balance weight is a general technique for all RLVR tasks. But if there exists some inherent correlation between response length and final label in some tasks, this technique could be useful to mitigate the bias introduced by different token weights.

---

> ### Author Response · Authors · 2025-12-03
> **Reply 5/n: Other details**
>
> 9.  **Q1, Q2.1**: Learning curve? inter-annotator agreement? disagremment ratio? incorrect proof case study?
>     - Yes, We test on the checkpoint and provide the accuracy curve in Appendix A.6. The curve shows the trend of improvement with more training data and RL steps.
>     - To get the label of our testset, we have two human annotators to label each proof independently. If the two annotators disagree, a third annotator will check the proof and make the final decision. In our testset with 238 samples, the inter-disagreement rate is 19%.
>     - We find that the LLM can roughly find the correct problem in the proof but sometimes could be confused by some minor problems. For example, in some cases, the LLM shows too strict to allow some commonsense steps or minor gaps. For more details, please see our Point 5 in response to yi4p as a error type breakdown and some case studies in Appendix A.9.
>
> 10. **Q2.2**, why we choose to train an ORM instead of a PRM:
>     - See point 2.
> 11. **Q3**, LLM-as-RM-for-RM:
>     - **Overhead**: Here we use an api calling of Deepseek-v3.1-chat because we find that check the fluency and some basic logic flow is not a hard task for LLMs. The specific overhead depends on the state of the server and how many GPUs you use for rollout. Notice that we use a one step delay strategy to avoid the waiting time of iterative rollout and training. This strategy is that when some nodes only generate rollout results and others only update the model. With this strategy, the api calling and reward calculation can be done in parallel with rollout generation or model training. In our training setting, where we use 48 GPU to rollout and 32 GPU to train, the time cost of LLM-as-a-RM-of-RM introduce 20% time overhead. With some most advanced parallel techniques introduced recently like ROLL[7], the overhead can be further reduced because the reward calculation can be totally parallelized with rollout generation and model training. The cost is roughly 0.5 dollar per step and we use about 200 dollar for our ~400 RL steps.
> 12. Some detailed analysis about our final performance. Which error types learned better? Some case studies? Some interesting examples?
>     - See Point 9.
>
> [1] Dekoninck et. al., The Open Proof Corpus: A Large-Scale Study of LLM-Generated Mathematical Proofs, 2025.
>
> [2] Ma et. al., Reliable Fine-Grained Evaluation of Natural Language Math Proofs, 2025.
>
> [3] Khatri et. al., The Art of Scaling Reinforcement Learning Compute for LLMs, 2025.
>
> [4] Wu et. al., Autoformalization with Large Language Models, 2022.
>
> [5] Azerbayev et. al., ProofNet: Autoformalizing and Formally Proving Undergraduate-Level Mathematics, 2023.
>
> [6] Poiroux et. al., Reliable Evaluation and Benchmarks for Statement Autoformalization, 2025.
>
> [7] Lu et. al., Part II: ROLL Flash – Accelerating RLVR and Agentic Training
> with Asynchrony, 2025.

---

### Official Review · Reviewer_yi4p · 2025-11-05

**Soundness:** 4
**Presentation:** 3
**Contribution:** 3
**Rating:** 8
**Confidence:** 4

**Summary:**

This paper addresses a critical bottleneck in advancing Large Language Models (LLMs) for high-level mathematical reasoning: the lack of a scalable and accurate Reward Model (RM) capable of evaluating full, proof-based solutions. While current LLM successes rely on Reinforcement Learning with Verifiable Rewards (RLVR) through simple answer matching, this approach fails for proofs, where the "asymmetry of verification" diminishes.

**Strengths:**

1. The multi-dimensional QPC data pipeline is highly innovative, leveraging the generation power of LLMs to create diverse (in style, length, and error type) data, while utilizing a clever hierarchical human check to ensure annotation accuracy and save vast amounts of expert time.
2. The paper insightfully diagnoses why current RLVR methods are insufficient for proofs and correctly pivots the task to training a verifier/RM first.
3. The proposed fixes for RL instability, the "LLM-as-RM for RM" for thought quality and the balanced token weight strategy for sequence length variation in order to provide highly practical and significant guidance for future work on generative reward models.
4.  Explicit efforts to ensure the Proof RM learns from diverse error types (Table 3 details how methods like "Mask Completion" induce more subtle, human-like errors) suggests a strong focus on generalizable verification beyond superficial checks.

**Weaknesses:**

1. While the composition-level human check is crucial for scalability, its efficacy depends entirely on the homogeneity of the data within a "composition" slice. The paper could be strengthened by providing more transparency on the statistical risks: if an entire slice is dropped due to a few non-conforming samples, high-quality data might be lost, or conversely, if the sample is not representative, a bad slice might be retained.
2. The fluency check performed by the "LLM-as-RM for RM" appears to only detect syntactic or surface-level issues (e.g., repeated sentences, nonsense) to prevent model collapse. It does not seem to penalize a coherent, yet logically flawed thought process (CoT) that still reaches the correct T/F conclusion. This could allow for subtle reward hacking in the reasoning process itself.

**Questions:**

1. Could you provide more details on the practical application of the composition-level check? Specifically, how many samples (and what proportion) were checked per composition, and what was the final rejection rate for entire compositions versus individual samples?
2. The RM is generative. Did you explore using an internal reward signal extracted from the RM's step-by-step reasoning (e.g., using a Process Reward Model (PRM) concept) in addition to the binary T/F (Outcome Reward Model) to better supervise the fluency and correctness of the intermediate steps, rather than just the final verdict?
3. Given the multi-dimensional diversity generation (Table 3), what is the breakdown of error types (e.g., case-analysis, hallucination, subtle step-level flaws) in the final training dataset? How does the ProofRM's accuracy vary across these different categories of errors compared to baselines?

---

> ### Author Response · Authors · 2025-12-02
> **Reply 1/n: providing more transparency on the statistical risks**
>
> We are glad to hear your positive feedback of our paper. We believe our data pipeline, dataset and the training recipe can provide practical reference of the field of the LLM post-training and reasoning. Here, we try to answer your questions as follows:
>
> 1. **W1**, providing more transparency on the **statistical risks**:
>     - Thank you for your reminder. The risk is theoretically possible but we have tried our best to avoid the risk, which is the key to make sure our data is both scalable and high-quality. Specifically,
>     - 1) Our **combination-level check** minimizes the bias between samples *within* a combination. The same dataset source means similar difficulty and problems style. The same extended LLM means similar knowledge, proof style and ability. And the same prompt method means similar generated proof behavior and error types. Therefore, within the combination, the samples have more similarity than those from other combinations. It means it is less likely to have the risk of our sampled check cannot represent the total batch.
>    - 2) **Confidence interval**: Our human check use a threshold 90% to make sure the final data quality is high enough. Considering select a question is a Bernoulli process, the *''overall''* accuracy on all data has a 95% confidence interval as $[p-1.96\sqrt{p(1-p)/n}, p+1.96\sqrt{p(1-p)/n}]$, where p is the accuracy on the checked samples $p\geq 90%$ and $n$ is the number of checked samples for this combination. Our $n$ is at least 30 and much larger than it for some larger combinations. So the lower bound of the 95% confidence interval is at least 79.3%. And the 90% confidence interval is $[p-1.645\sqrt{p(1-p)/n}, p+1.645\sqrt{p(1-p)/n}]$, where the lower bound is at least 83%. It means that we highly confident that the full data accuracy is at least 83%, which is an acceptable quality for large-scaling RL training.
>    - 3) Above, we make sure we only keep the samples with high performance. On the other hand, there is the risk we may indeed have been too strict in filtering out some correct data. But we believe it is more important to make sure the samples we keep are all high-quality. As some previous work pointed out, the accuracy of reward is important to make sure the RL training is stable. At the same time, although data generation is expensive and it is better to not waste them, it is an automatic process and cheaper than human annotation one-by-one to avoid these risks.

---

> ### Author Response · Authors · 2025-12-02
> **Reply 2/n: reward hacking**
>
> 2. **W2**, **reward hacking**:
>    - Yes, we acknowledge that reward hacking is a basic drawback of RLVR, despite its practical success. RLVR can be considered as an automatic proxy of a verifier, where there is theoretically risk of reward hacking. In our paper, our method is to RLVR the *"first-order"* verifier (i.e., the verifier of the prover) instead of the solver itself, that is, we use the automatic proxy as the *"second-order"* verifier instead of the *"first-order"* verifier. Thanks to the *"asymmetric of verification"* in these tasks, the difficulty of higher-order verification is easier than the lower-order ones, so the risk of the reward hacking is smaller.
>    - During our practical training, we manually inspected rollout generations every few dozen steps. These spot-checks did not reveal a consistent reward-hacking pattern of "fluent language with incorrect reasoning". While manual inspection cannot rule out rare or subtle failures, it suggests such behavior was not a dominant or systematic mode during optimization.
>     - To further solve the problem, we have an idea for future work about *"chain-of-verifying"*. The core idea is to reduce verification difficulty by introducing additional verification over intermediate verification artifacts. For example, although verifying proof correctness is a global decision, a verifier can emit a structured, localized artifact (e.g., a list of suspected faulty steps with pointers), and then a second-stage checker only needs to validate these localized claims. We will present and evaluate this iterative verification-of-verification framework in future work. Through this framework, we can not only check the outcome but also check the *"check process"* to further reduce the risk of reward hacking.
>
> p.s: Here, we are glad to provide a discussion about why in our setting, the surface-level issues could be more ''important''. The key difference is **strength** of the correlation between the process correctness and final correctness. The surface-level issue is more **independent** with the reasoning of LLM and could be somehow ignored by the model at their first occurrence, which incorrectly encouraged and finally degrade the model's behavior. However, the wrong logic flow is more related to the reasoning results and directly harm the final correctness at their first occurrence. So it is less likely to be incorrectly encouraged and harm the model's behavior.

---

> ### Author Response · Authors · 2025-12-02
> **Reply 3/n: Details of the composition-level check & PRM**
>
> 3. Q1, **Details of the composition-level check**:
>    - Sure, our details of the combination-level check is here. As a summary, for each combination of (dataset source, extended LLM, prompt method), we randomly sample at least 30 generated proofs and have human check their correctness. If the accuracy is lower than 90%, we discard all samples with this combination. Otherwise, we keep all samples with this combination. Among 30 combinations, we reject 12 of them. As for individual samples, because we only generate full data for the combinations passing the check, we cannot count the number of reject samples. We estimate the reject ratio is about 40% based on the checked samples. We also update the details in Appendix A.4.6.
> 4. Q2, **PRM**:
>    - As you mention, our data pipeline can be easily extended to PRM. But to get accurate PRM is still a challenging problem for its nature. For example, in some problems, the value of a reasoning step cannot be determined until the final proof is finished. If a latter step is wrong, we cannot determine whether the current step is valuable or not. We are now exploring ways to get better PRM (one possible way is the "chain-of-verifying" mentioned above) but it is far from the scope of this paper.

---

> ### Author Response · Authors · 2025-12-02
> **Reply 4/n: breakdown of error types and detailed performance analysis**
>
> 5. Q3.1, **breakdown of error types**:
>    - **Collect a subset to breakdown**: To quantify the distribution of error types, we sampled the data and conducted manual annotations. We randomly sampled 97 QA pairs in our training set with labeling `False`. For each pair, we had human annotators identify the primary error type in the generated proof.
>    - **Error type labeling**: For a logic deduction, we roughly model a logic step as a syllogism. So we categorize the errors based on the three key elements of a syllogism: *premises*, *logic flow*, and *conclusion.* The first error type corresponds to issues with premises, the second to issues with logic flow, and the third to issues with conclusions. We find most errors are due to incomplete logic flow, which indicates that the model fails to chain the reasoning steps rigorously.
>    - **Results**: The results are summarized in the table below:
>
>        | Error Type                          | Percentage   |  ProofRM Accurate | Base Model Accurate | Improvement |
>        |-------------------------------------|--------------|-------------------|---------------------|----|
>        | Misusing unjustified assumptions    | 15.5%        | 61.7%               | 53.3%                 | 8.4% |
>        | Incomplete logic flow               | 56.7%        | 56.8%               | 42.7%                 | 14.1% |
>        | Making up non-existent conclusions  | 27.8%        | 55.6%               | 41.3%                 | 14.3% |
>
>    - **Analysis**: Let's analyze the results in the table. Notice that the performance of base model is even lower than random guess (50%), suggesting the model easily believes incorrect proofs. ProofRM improves accuracy across these error types, with higher gains in the second and third categories, where the base models shows weaker performance. It shows that our RL stage indeed helps our model to avoid the easy-believing behavior and find these errors.
>    - **Case study**: We have added some case study in Appendix A.9 to support this claim.

---

### Meta-Review · Area_Chair_Qx9h · 2026-01-07

**Summary:**

Here is a summary of the reviewers' concerns.

 - Slice-level quality control may still be brittle. Accepting/rejecting whole “compositions” from small audits can drop good data or keep bad slices. Still unclear how representative slices are in practice and how much instance-level noise remains. (yi4p, WK1o, S5u1)
 - Process-faithfulness / reward hacking risk remains. The auxiliary “LLM-as-RM-for-RM” mainly targets surface fluency. It may not reliably punish coherent-but-wrong reasoning traces, so optimization could still exploit proxy signals instead of true logical validity. (yi4p, S5u1)
 - Evaluation credibility / statistical strength. Key test-time scaling uses few problems (even if many proofs), plus some curation/cleanup. Reviewers still worry about confidence intervals, significance, and whether the best-of-k results are robust rather than benchmark-dependent. (WK1o, S5u1, BDhi)
 - Generalization gaps. Large drop on student-style proofs and limited coverage of truly out-of-distribution proof styles. No convincing evaluation on research-level math. Formal proof domains are excluded (whether that’s acceptable or not). (WK1o, S5u1)
 - Baselines and methodological positioning. Despite added comparisons/ablations, concerns linger that this is primarily an engineering consolidation. Limited head-to-head against strong proof-verification baselines and limited exploration of PRM/search-based alternatives. (WK1o, S5u1, BDhi)
 - Ranking/calibration vs pointwise accuracy. The model can classify single proofs but may be weak at relative ranking needed for best-of-k. The “bug / trivial negatives” explanation doesn’t fully resolve whether binary supervision yields poor calibration. (BDhi, S5u1)
 - Compute/efficiency and practicality. Generative verification with repeated sampling can be expensive. Time/throughput comparisons against scalar/discriminative RMs and the true cost of the auxiliary LLM filter remain a sticking point for deployment realism. (WK1o, S5u1)

**Reviewer Concerns:**

Principal concerns that remain after rebuttal.

 - Evaluation still not fully convincing. Too few distinct problems, some curation/cleanup, limited significance reporting, and mixed OOD coverage. (WK1o, S5u1, BDhi)
 - Generalization is weak where it matters most. Large absolute drop on student-style proofs, no solid research-level math test, OOD gains often small. (WK1o, S5u1)
 - Process-faithfulness is unresolved. Current safeguards mainly catch surface failures, coherent-but-wrong reasoning (and reward hacking) is not rigorously ruled out. (yi4p, S5u1)
 - Ranking/calibration remains a principal gap. Strong pointwise T/F accuracy does not reliably translate into strong best-of-k selection. (BDhi, S5u1)
 - Comparative positioning is incomplete. Limited head-to-head against strong proof-verification baselines and limited empirical study of PRM/search-based alternatives. (WK1o, S5u1, BDhi)
 - Practical cost is still uncertain. Generative verification + repeated sampling may be expensive, efficiency comparisons are not standardized across baselines. (WK1o, S5u1)

**Reviewer Scores:**

In the view of remaining issues, the scores would most likely have remained unchanged

---

### Decision · Program_Chairs · 2026-01-26

Reject